# Low-temperature anode-free potassium metal batteries

Mengyao Tang ©[1,3], Shuai Dong[1,3], Jiawei Wang[1], Liwei Cheng[1], Qiaonan Zhu[1], Yanmei Li[2], Xiuyi Yang[1], Lin Guo ©[1] ✉ & Hua Wang ©[1] ✉

In contrast to conventional batteries, anode-free configurations can extend cell-level energy densities closer to the theoretical limit. However, realizing alkali metal plating/stripping on a bare current collector with high reversibility is challenging, especially at low temperature, as an unstable solid-electrolyte interphase and uncontrolled dendrite growth occur more easily. Here, a low-temperature anode-free potassium (K) metal non-aqueous battery is reported. By introducing Si-O-based additives, namely polydimethylsiloxane, in a weak-solvation low-concentration electrolyte of 0.4 M potassium hexafluorophosphate in 1,2-dimethoxyethane, the in situ formed potassiophilic interface enables uniform K deposition, and offers K||Cu cells with an average K plating/stripping Coulombic efficiency of 99.80% at −40 °C. Consequently, anode-free Cu||prepotassiated 3,4,9,10-perylene-tetracarboxylicacid-dianhydride full batteries achieve stable cycling with a high specific energy of 152 Wh kg$^{-1}$ based on the total mass of the negative and positive electrodes at 0.2 C (26 mA g$^{-1}$) charge/discharge and −40 °C.

Currently, high-energy alkali metal batteries are being intensively pursued to meet the ever-increasing requirements for energy storage in modern society[1]. With the goal of maximizing the gravimetric and volumetric energy densities, intensive attention is being placed on anode-free cells with zero excess alkali metals[2]. Nevertheless, their implementation at low temperature is challenged by the low reversibility, unstable solid-electrolyte interphase (SEI) and uncontrolled dendrite growth of alkali metal at the anode side. On the one hand, metal nuclei deposited with a reduced size at low temperature are more likely to be exposed to the electrolyte[3], which aggravates the metal loss and causes a poor Coulombic efficiency (CE). On the other hand, the formation of passivating ingredients in the SEI is restricted due to insufficient low-temperature dynamics[4], leading to incomplete anode passivation and persistent side reactions, which are also reflected in the low CE. Recently, extensive efforts have been devoted to improving the reversibility of alkali metal anodes at low temperatures, and several strategies have been employed, including (1) modifying current collectors to induce the formation of enhanced

passivation layers[4]; (2) employing novel solvents, such as liquefied gases, to improve their reduction resistance against metal anodes[5,6]; and (3) introducing additives to generate a stable SEI and suppress dendrite growth[7]. Unfortunately, the reported CE of alkali metal anodes is far from satisfactory and barely breaks 99% below −20 °C. In this case, a large excess of alkali metal is needed to compensate for irreversible consumption during cycling, which greatly compromises the energy density of batteries. Therefore, ameliorating the reversibility of alkali metals is critical for the realization of low-temperature anode-free alkali metal batteries.

Although lithium metal batteries have been attracting extensive attention for low-temperature applications, the possibility of a better alternative should be further explored. Generally, the interactions of charge carriers (Li$^+$, Na$^+$, K$^+$) with organic solvents play a crucial role in determining the performance of alkali metal batteries[8,9]. Owing to the low Lewis acidity, K$^+$ ions have weak Coulombic interactions with solvent molecules and would exhibit small Stokes radii, which can effectively reduce the ion diffusion resistance in electrolyte solutions

[1]School of Chemistry, Key Laboratory of Bio-Inspired Smart Interfacial Science and Technology of Ministry of Education, Beihang University, Beijing, China. [2]School of Materials Science and Engineering, University of Science and Technology Beijing, Beijing, China. [3]These authors contributed equally: Mengyao Tang, Shuai Dong. ✉e-mail: guolin@buaa.edu.cn; wanghua8651@buaa.edu.cn

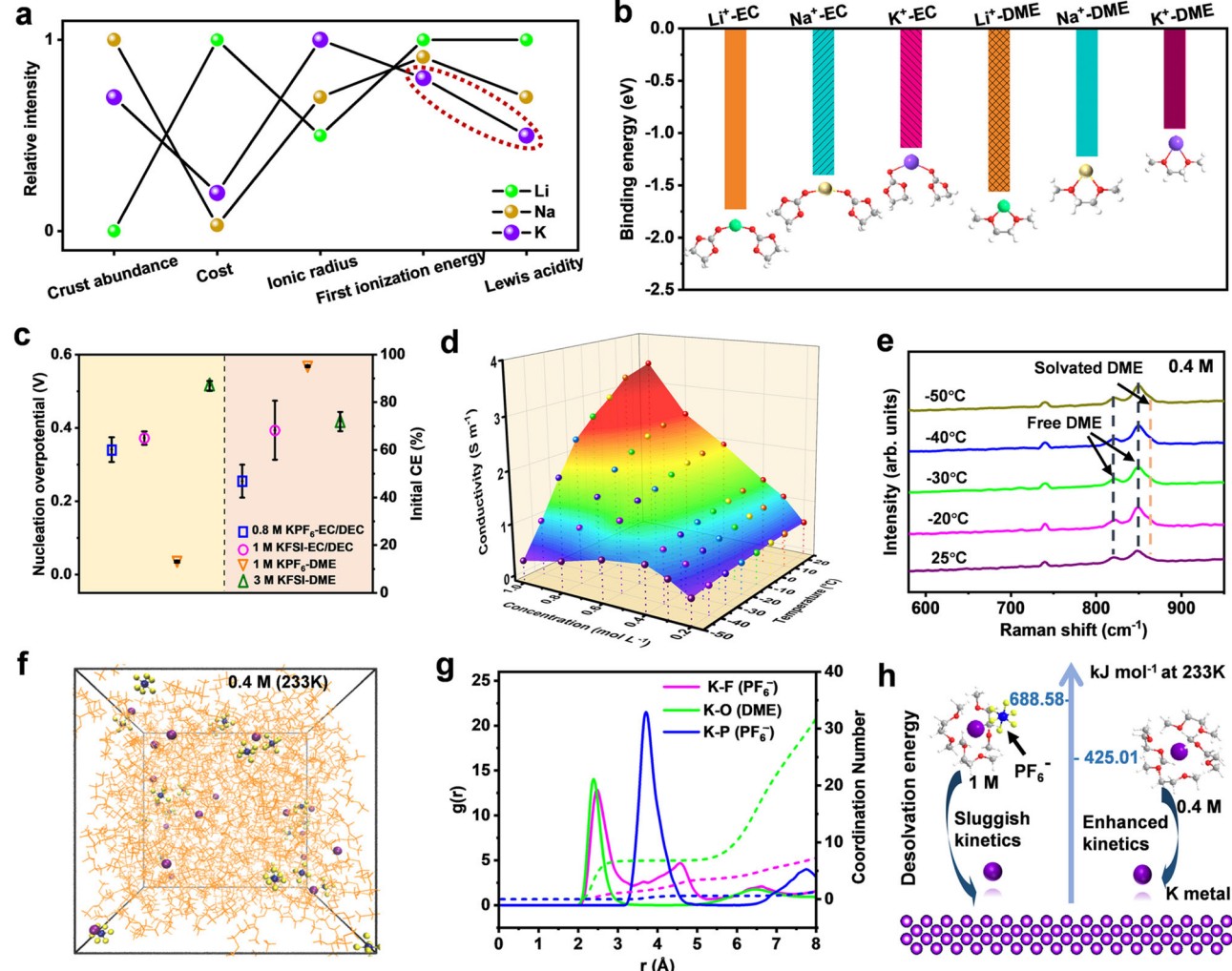

**Fig. 1 | Characterization of 0.4 M KPF₆-DME at low temperatures.**
**a** Physicochemical properties of Li, Na and K elements. **b** Binding energies of solvent·K⁺/Na⁺/Li⁺ complexes. **c** Averaged initial CE and nucleation overpotential values of K||Cu cells with four conventional electrolytes at 1 mA cm⁻²/1 mAh cm⁻² and 25 °C. Standard errors are 0.8 M KPF₆-EC/DEC (0.341 ± 0.033 V and 46.89 ± 6.91%), 1 M KFSI-EC/DEC (0.372 ± 0.018 V and 68.26 ± 12.36%), 1 M KPF₆-DME (0.035 ± 0.003 V and 95.12 ± 0.35%) and 3 M KFSI-DME (0.514 ± 0.013 V and 71.89 ± 4.04%) with three identical samples. **d** Temperature-dependent ionic conductivity of KPF₆-DME electrolytes with a series of concentration gradients from 0.2 to 1.0 M. **e** In situ temperature-dependent Raman spectroscopy of 0.4 M KPF₆-DME. **f** Snapshots of the MD simulation boxes of 0.4 M KPF₆-DME at 233 K (−40 °C). **g** The corresponding RDF data. **h**, Calculated desolvation energies in 0.4 and 1 M KPF₆-DME at 233 K. Li, Na, K, C, O, H, F and P are marked with green, gold, purple, grey, red, white, yellow, and blue, respectively.

(Fig. 1a and Supplementary Table 1)[10]. As hypothesized, in typical ether-based (1,2-dimethoxyethane (DME)) and ester-based solvents (ethylene carbonate (EC)), the calculated binding energies of solvent·K⁺ complexes are lower than those of Na⁺ and Li⁺, favouring fast ion diffusion (Fig. 1b). For this reason, K-based electrolytes are more likely to mitigate the negative effect of low temperature on ionic conductivity. To date, a few studies have demonstrated the feasibility of K metal anodes cycled at low temperature[11–13]. However, the reported K metal anode usually operates with a large overpotential and low current density. More importantly, a high CE (>99%), as a performance goalpost for K metal anodes, has never been realized at low temperature due to their high reactivity (i.e., the low first ionization energy) to solvent, unstable SEI and intractable K dendrite growth[14]. Therefore, the realization of highly reversible and stable K metal plating/stripping behaviour at low temperatures still involves great challenges.

Here, a low-temperature anode-free K metal battery was first achieved by adjusting the electrolyte chemistry. The low-concentration KPF₆/DME electrolyte exhibits a high ionic conductivity and weak K⁺ solvation effect at low temperature. Moreover, polydimethylsiloxane capped with −OCH₃ groups (PDMS), as an electrolyte additive, can

graft on the K metal surface and induce the formation of Si-O-K and KF-based robust organic–inorganic hybrid SEI, which is critical to the reversibility and cycling stability of K metal at low temperature. Consequently, inspired by the highly reversible K plating/stripping behaviour, the anode-free Cu||prepotassiated 3,4,9,10-perylene-tetracarboxylicacid-dianhydride (KPTCDA) full cell was assembled and exhibits a high specific energy of 152 Wh kg⁻¹ based on the full mass of the negative and positive electrodes at 0.2 C and −40 °C. This result may spur the potential application of anode-free K metal batteries in powering unmanned aerial vehicles and rovers for the polar exploration.

## Results

### Optimizing low-temperature electrolyte systems
Conventional electrolytes for K⁺ ion batteries[15,16], including 0.8 M potassium hexafluorophosphate (KPF₆) in EC/diethylene carbonate (DEC; 1:1 v/v), 1 M KPF₆ in DME, 1 M potassium bis(fluorosulfonyl)imide (KFSI) in EC/DEC and 3 M KFSI in DME, were screened for compatibility with the K metal anode. The KPF₆/DME-based electrolyte system in K|| Cu cells exhibits distinct advantages in terms of higher initial CE, lower

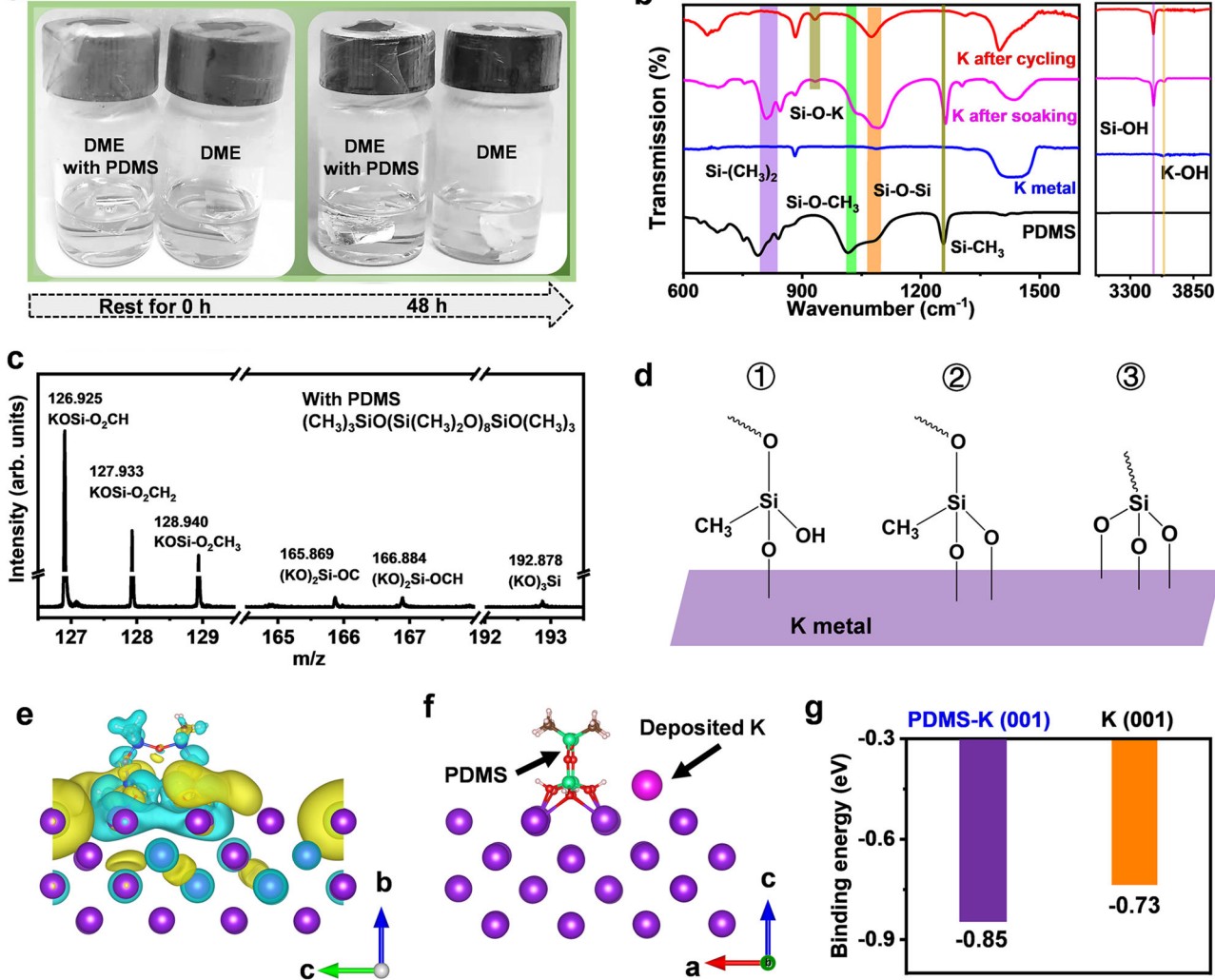

**Fig. 2 | Investigation of the effect of the PDMS additive on K anode. a** Digital photographs of K soaked in DME solvent with and without 2 vol. % PDMS for 48 h. **b** ATR-FTIR spectra of K metal treated by soaking in PDMS for 5 min or cycling in K|| Cu cells with 2 vol. % PDMS for 5 cycles. **c** Selected TOF-SIMS ion spectra of the cycled K anode with 2 vol. % PDMS. **d** Schematic illustration of binding modes of PDMS-K. **e** Differential charge densities for the most stable adsorption configuration of PDMS-K. Yellow is the electron-accumulation area, while cyan represents the electron-depletion area. **f** Structural model of one K adatom on PDMS-K (001) substrate. **g** DFT-calculated binding energies of the K adatom on PDMS-K (001) and bare K (001) substrates, respectively. The colour codes are O (red), C (grey), H (white), Si (green), and K (purple), respectively.

nucleation overpotential and well cold-adapted property (Fig. 1c and Supplementary Figs. 1 and 2). Furthermore, at a given temperature, there is usually an optimal electrolyte concentration that results in the highest ionic conductivity[17–19]. As shown in Fig. 1d and Supplementary Fig. 3, 0.4 M KPF$_6$-DME exhibits a high ionic conduction of 0.89 and 0.80 S/m at ultralow temperatures of −40 and −50 °C, respectively. In addition, the peaks of free DME and solvated DME show no obvious change according to in situ temperature-dependent Raman spectra[20], indicating the excellent stability of this 0.4 M electrolyte for low-temperature applications (Fig. 1e). To gain a deeper understanding of the solvation structure at −40 °C, molecular dynamics (MD) simulations were carried out[21,22]. In 0.4 M KPF$_6$-DME, K$^+$ ions are tightly solvated by DME molecules, and the coordination number of K-O$_{DME}$ in the radial distribution functions (RDFs) was calculated to be 6.78, referring to the representative K$^+$-(DME)$_4$ complex (Fig. 1f, g). While in 1 M KPF$_6$-DME, the PF$_6^-$ anion has a higher possibility of entering the first K$^+$ solvation sheath with a coordination number of 0.94, corresponding to the K$^+$-(DME)$_3$-PF$_6^-$ complex (Supplementary Fig. 4 and Supplementary Table 2). Then, the intensified association of cation–anion (K$^+$-PF$_6^-$) was confirmed by Raman spectra of KPF$_6$/DME

electrolytes as the salt concentration increased (Supplementary Fig. 5). Based on these proposed solvation structures, the calculated K$^+$ ion desolvation energy in 0.4 M KPF$_6$-DME (425.01 eV) is much lower than that in the 1 M KPF$_6$-DME electrolyte (688.58 eV) (Fig. 1h). Meanwhile, the decreased energy barrier for K$^+$ ion desolvation is further demonstrated by the electrochemical impedance spectroscopy (EIS), in which 0.4 M KPF$_6$-DME exhibits a relatively low charge-transfer resistance (Rct) (Supplementary Fig. 6 and Supplementary Table 3). Overall, this 0.4 M KPF$_6$-DME electrolyte exhibits weak solvation ability and high ionic conductivity at low temperature.

**Regulating K deposition behaviour by the PDMS electrolyte additive**

In addition to the enhanced ion transport kinetics, constructing a stable metal/electrolyte interface is also crucial for a highly reversible K metal anode at low temperature. Unexpectedly, the anhydrous DME solvent becomes cloudy after K soaking for 48 h (Fig. 2a and Supplementary Fig. 7). Then, the decomposition product of DME molecules on the K metal surface was clearly detected by attenuated total reflection-Fourier transform infrared (ATR-FTIR), and it could be

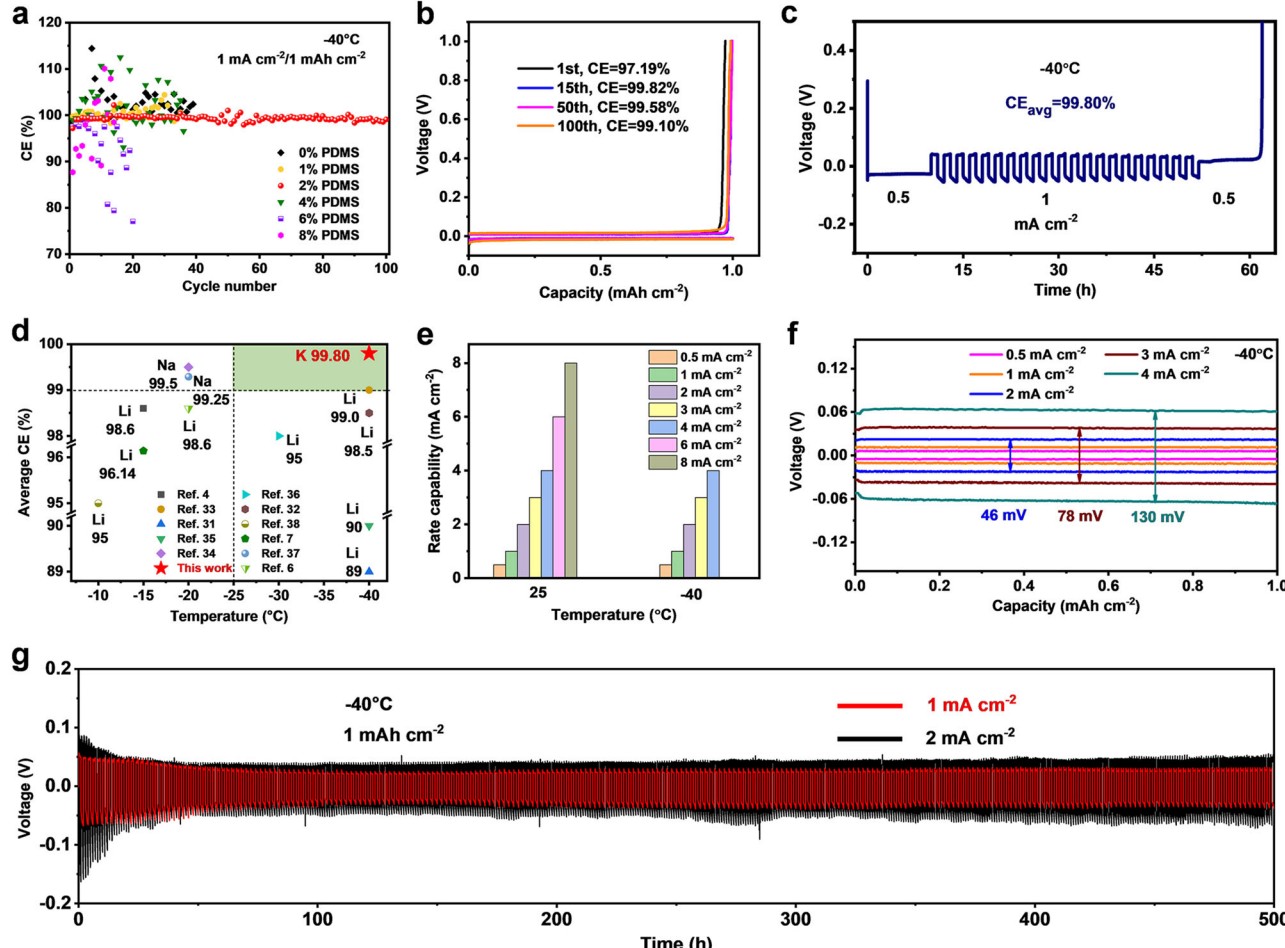

**Fig. 3 | Electrochemical characterizations of K metal anodes at low temperatures. a** K plating/stripping CE of K‖Cu half cells with various electrolytes at 1 mA cm⁻²/1 mAh cm⁻² and −40 °C. **b** Corresponding plating/stripping curves at different cycles with the KDP electrolyte. **c** Average CE determined by the Aurbach method. **d** Comparison of CE values among relevant low-temperature alkali metal anodes. **e** Rate capability of the K‖K cells with the KDP electrolyte at 25 and −40 °C, respectively. **f** Representative voltage profiles of K‖K cells with the KDP electrolyte at different current densities and −40 °C. **g** Cycling performance of K‖K cells with the KDP electrolyte at 1 mA cm⁻²/1 mAh cm⁻² and 2 mA cm⁻²/1 mAh cm⁻².

identified as $CH_3COO^-$ (Supplementary Fig. 8)[23]. In sharp contrast, the DME solvent with the addition of the 2 vol. % PDMS additive is still clear, and the K surface with reduced $CH_3COO^-$ accumulation remains bright, indicating that PDMS can efficiently suppress the continuous decomposition of DME on the K metal surface. To explore the interaction mechanism between K and PDMS, ATR-FTIR and time-of-flight secondary ion mass spectrometry (TOF-SIMS) were carried out (Fig. 2b, c). It can be found that the spontaneous hydrolytic condensation reaction between Si-$(OCH_3)_3$ (1016 cm⁻¹) and KOH (3587 cm⁻¹) occurs after K soaking in DME with PDMS additive, then PDMS molecules graft on the K metal surface by the formed K-O-Si chemical bonds (933 cm⁻¹)[24,25]. In addition, Si-$CH_3$ (1261 cm⁻¹) reacts to form Si-OH (3496 cm⁻¹), and PDMS can be crosslinked with neighbours by Si-O-Si bonds to form polysiloxane, which is facilitated by an electric field (Supplementary Fig. 9)[26]. Moreover, ion fragments of KOSi-$O_2CH_3$, $(KO)_2$Si-OCH, and $(KO)_3$Si are detected at mass-to-charge ratios (m/z) of 128.940, 166.884, and 192.878, respectively, illustrating that PDMS molecules and the K metal substrate are linked by $(K-O)_n$-Si (n = 1, 2, 3) chemical bonds (Fig. 2d)[27]. Based on the above analysis, a PDMS protective layer is formed in situ on the K metal surface, which is essential for the interfacial stability of K metal anodes. To gain atomic-level insights into the effect of PDMS on the deposition behaviour of K⁺ ions, density functional theory (DFT) calculations including differential charge densities and binding energies were conducted[28,29]. The results

reveal that PDMS alters the electric field environment of K metal surface by Si-O and forms electron-rich nucleophilic regions (marked with yellow colour, as shown in Fig. 2e). Compared to that of the bare K (001) substrate, the binding energy of the K atom with PDMS-K (001) shows a lower value of −0.85 eV, thus favouring the absorption of the K atom on PDMS-K surfaces (Fig. 2f, g and Supplementary Fig. 10). Therefore, PDMS can help improve the interfacial stability and serve as a potassiophilic interface to guide uniform K deposition.

## Realizing highly reversible K metal plating/stripping at low temperature

Then, the K⁺ plating/stripping behaviour in 0.4 M $KPF_6$-DME electrolytes with different volume fractions of PDMS additive was evaluated −40 °C (Fig. 3a and Supplementary Fig. 11). It can be found that 0.4 M $KPF_6$-DME with 2 vol. % PDMS additive (referred to as the KDP electrolyte) shows excellent compatibility with the K metal anode during cycling, in which the CE increases from an initial 97.19% to 99.82% after 15 cycle and remains at 99.10% even after 100 cycles (Fig. 3b and Supplementary Fig. 12). In addition, the reversibility of K anodes over a wide range of current densities was investigated in K‖Cu cells at −40 °C. An average CE of 99.84% is observed at 0.5 mA cm⁻², and even at a higher current density of 2 mA cm⁻², the cell cycles well with a stable CE of 98.95%. When brought back to 0.5 mA cm⁻², K‖Cu cells with a CE marginally higher than 100% are observed at the beginning of

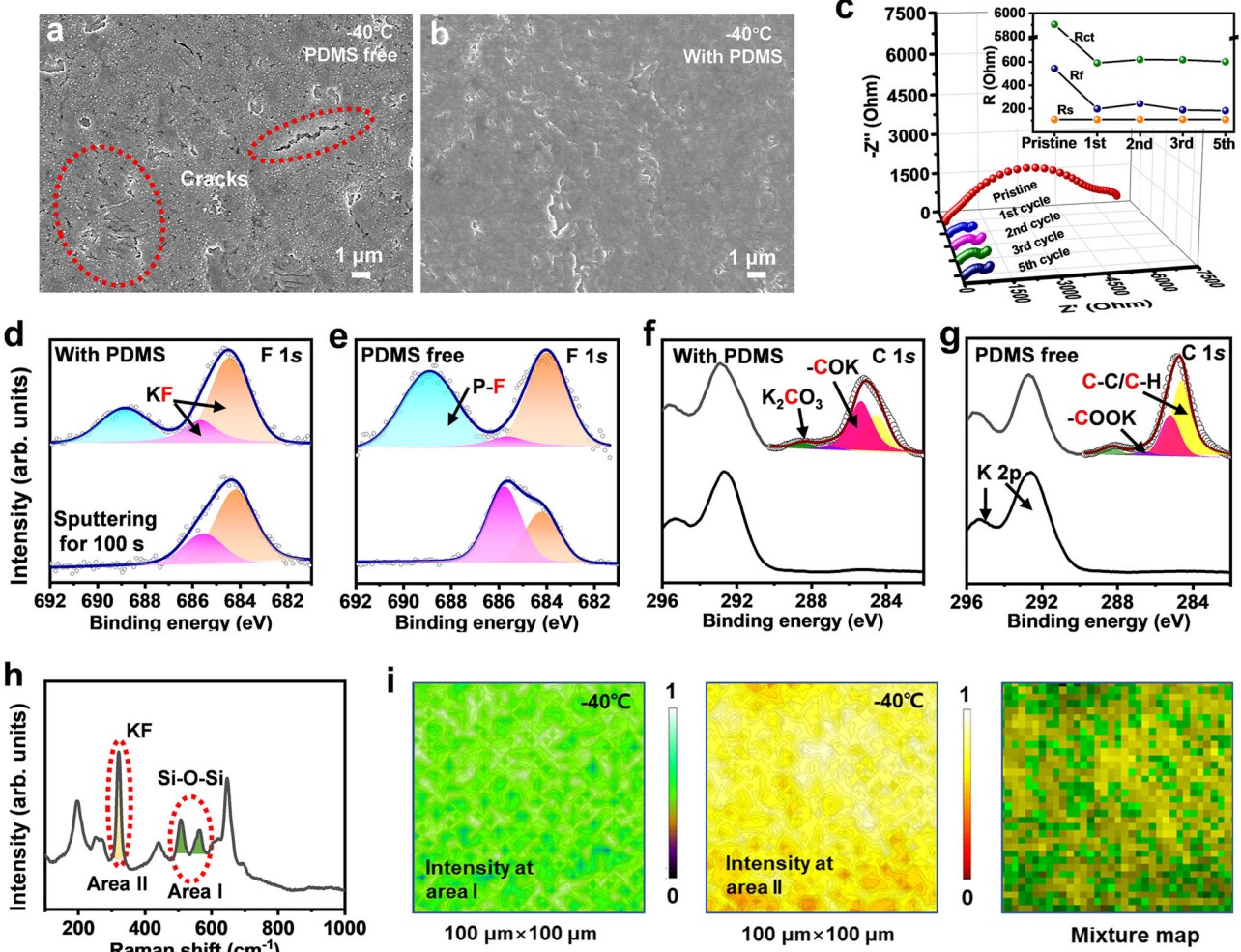

**Fig. 4 | The ingredient analysis of SEI and its spatial distribution.** Top-view SEM images of the K deposit layer on Cu with **a** 0.4 M KPF$_6$-DME and **b** the KDP electrolyte at −40 °C (1.0 mA cm$^{-2}$/1 mAh cm$^{-2}$). **c** Nyquist plots and quantitative analysis of EIS in K∥Cu cells cycled at −40 °C with the KDP electrolyte. In-depth F 1$s$ and C 1$s$ XPS spectra of the K anode cycled in **d**, **f** KDP and **e**, **g** 0.4 M KPF$_6$-DME at −40 °C, respectively. **h** Raman spectra of the cycled K anode with the KDP electrolyte at −40 °C and **i** corresponding Raman mapping images.

cycles, indicating that dead K formed at higher current densities are recovered (Supplementary Fig. 13)[6]. As determined by the most common Aurbach method[30], the average CE of K metal in K∥Cu asymmetric cells gains 99.80% (Fig. 3c); thus, an highly reversible K metal anode has been realized at low temperature. Encouragingly, this 99.80% CE is well-placed among all low-temperature (<0 °C) alkali metal (Li, Na, K) anodes reported thus far (Fig. 3d)[4,6,7,31–38]. Moreover, K∥K cells with the KDP electrolyte can stably cycle over a temperature range from 25 °C to −50 °C, and the sudden increase in the overvoltage at −60 °C is due to phase transition of the electrolyte, which is identified by the observable endothermic peak arising at approximately −54 °C through differential scanning calorimetry (DSC) (Supplementary Figs. 14 and 15)[9]. More importantly, the KDP electrolyte maintains a high ionic conductivity of 0.69 S/m at −40 °C, which is conducive to desirable rate capability (Supplementary Fig. 16). As shown in Fig. 3e, the K∥K cells can cycle well with a stepwise increasing current density from 0.5 to 8 mA cm$^{-2}$ at 25 °C and even survive up to 4 mA cm$^{-2}$ at −40 °C. A relatively low overpotential of 130 mV at a high current density of 4 mA cm$^{-2}$ is achieved at −40 °C, only slightly higher than that of K∥K cells at 25 °C; thus, the KDP electrolyte is quite feasible for low-temperature applications (Fig. 3f and Supplementary Fig. 17). Without any activation pretreatment, K∥K cells can stably cycle without K dendrites over 500 h at 1 and 2 mA cm$^{-2}$, respectively, with a capacity of 1 mAh cm$^{-2}$ (Fig. 3g and Supplementary Fig. 18). A quantitative

comparison of the key parameters for the low-temperature K metal anode is summarized in Supplementary Table 4[11–13], and the state-of-the-art performance, including low electrochemical polarization, high-rate capability and large cumulative plated capacity (CPC) is achieved in our work. Inspiringly, the high reversibility of K metal plating/stripping behaviour with an average CE of 99.80% is expected to enable stable cycling of anode-free K metal full batteries.

## Robust SEI contributes to highly reversible K metal anodes

First, scanning electron microscopy (SEM) were performed to understand the influence of PDMS on the K plating morphology. A tense and uniform K deposit layer without dendrites on the Cu substrate is observed in the KDP electrolyte at −40 °C, whereas sharp cracks appear in the PDMS-free electrolyte owing to the unstable SEI and 'dead' K formation (Fig. 4a, b). Furthermore, the electrochemical impedance spectra (EIS) of K∥Cu cells were collected from the pristine state to the fifth cycle at −40 °C (Fig. 4c, Supplementary Fig. 19, and Supplementary Table 5). After the first-cycle activation, the Rct drops from the original 4872 to 337 Ω, but the bulk electrolyte resistance (Rs) is almost the same around 108 Ω. In the subsequent cycles, the impendence remains stable, benefitting the uniform deposition and reversible stripping of K during cycling[39]. The results further indicate that PDMS contributes to the formation of a stable SEI on the K anode. Then, the chemical composition of the SEI was investigated by X-ray

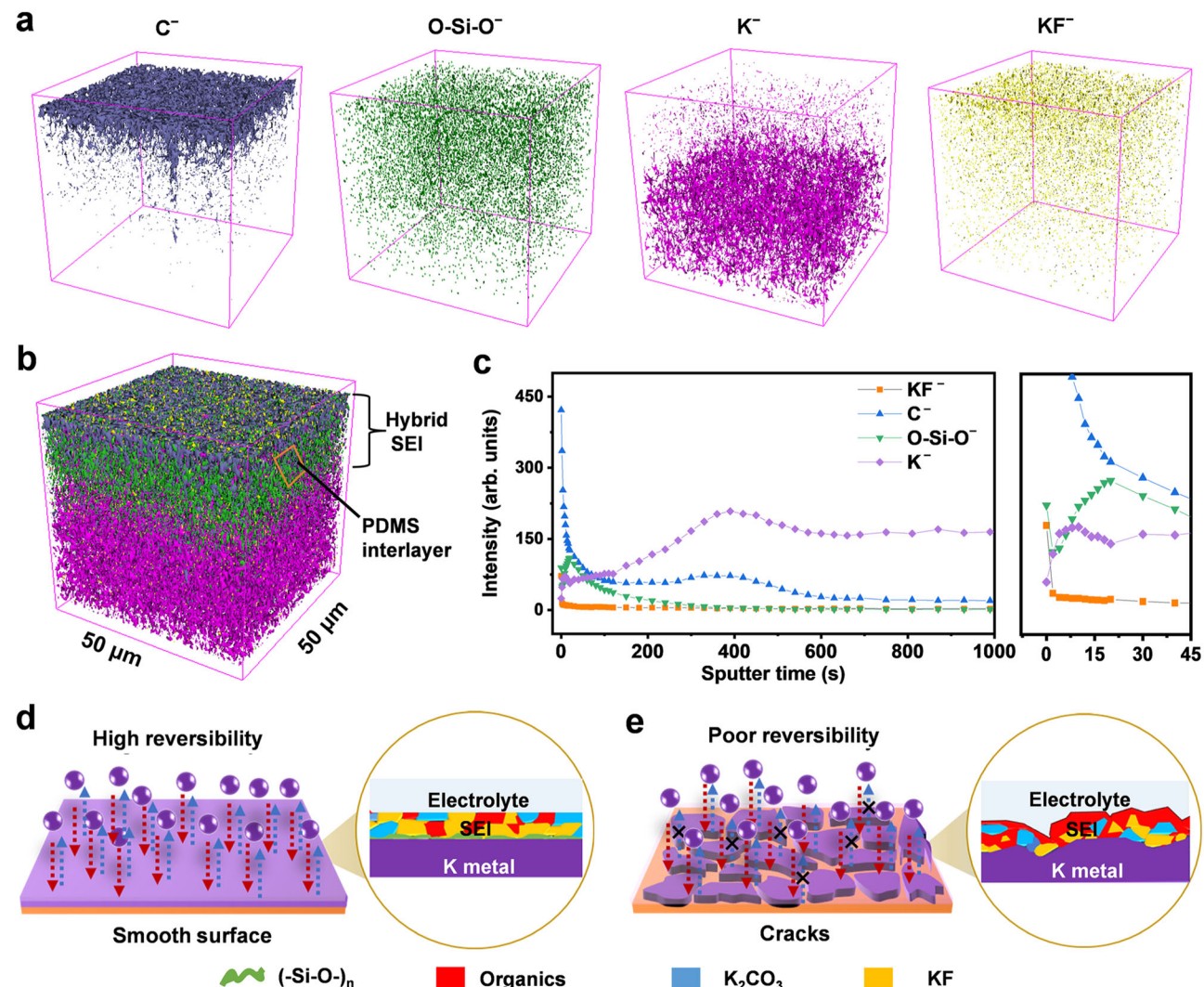

**Fig. 5 | TOF-SIMS 3D distribution analysis of K metal cycled with the KDP electrolyte at −40 °C. a** 3D distribution of related ion fragments. **b** 3D distribution overlay of (**a**). **c** Depth profile curves of TOF-SIMS in negative mode. The samples of K electrodeposits were prepared from K||Cu cells with 1 mA cm$^{-2}$/1 mAh cm$^{-2}$ at −40 °C. Schematic illustration of the SEI film derived from the 0.4 M KPF$_6$-DME electrolyte **d** with and **e** without 2 vol. % PDMS additives.

photoelectron spectroscopy (XPS). As depicted in Fig. 4d, e and Supplementary Fig. 20a, b, it is found that the outer SEI layer formed on the K anode with the KDP electrolyte shows a KF-rich phase (peaks at 684.6 and 685.7 eV in the F 1s spectrum), accounting for 12.5% and 15.2% of the total (C + F) contents at 25 and −40 °C, respectively, which is critical for the SEI stability[40]. Moreover, the C 1s signal decreases from 86.1% to 78.9% with the addition of PDMS at −40 °C, demonstrating that the organic species of the SEI originating from DME decomposition are reduced (Fig. 4f,g and Supplementary Fig. 21). Specifically, -COK (285.6 eV) noticeably increases, whereas the contents of K$_2$CO$_3$ (288.6 eV) and -COOK (286.9 eV) decrease. The same trend also occurs for the SEI formed at 25 °C (Supplementary Fig. 20c, d). These results suggest that the PDMS layer can prevent electrons on the K surface from continuously attacking C-O to form C = O, thus efficiently suppressing the chain reaction of DME decomposition[23,40–42]. After sputtering for 100 s (-10 nm) with Ar⁺, the inner SEI layer exhibits a KF-dominant phase regardless of the temperature and additive difference. Besides, the weak Si 2p XPS signal is enhanced, indicating that PDMS should be adjacent to the K surface and allow the K-ion flux to permeate (Supplementary Fig. 22)[27]. Furthermore, the Raman peaks in the frequency regions of 200-400 and 500-600 cm$^{-1}$ can be assigned to the vibration model of KF and the symmetric stretching vibrations of Si-O-Si, respectively (Fig. 4h)[43,44].

To generate a more representative and homogenous image of the SEI layer[45], a large area of 100 μm × 100 μm was scanned. The Raman signal-mapping images of Si-O-Si (PDMS) and KF shown in Fig. 4i are generated from their peak intensities, enabling the spatial distribution of KF and PDMS on the K surface to be clearly visualized. It can be found that KF and PDMS are uniformly distributed on the K metal surface, suggesting that a stable organic–inorganic hybrid interface layer is formed in the optimized electrolyte.

Furthermore, the three-dimensional (3D) distributions of these organic and inorganic species were reconstructed by TOF-SIMS, including C⁻, K⁻, KF⁻ and O-Si-O⁻ with m/z = 12.000, 59.967, 38.964 and 57.967, respectively (Fig. 5a). Interestingly, the 3D distribution overlay of these four ion fragments displays a sandwich structure with an O-Si-O⁻ interlayer, indicating that PDMS can work well as a protective layer on K metal surface (Fig. 5b). Based on the corresponding depth profile curves, the signal intensity of elemental C⁻ and KF⁻ from electrolyte decomposition sharply decreased, while that of O-Si-O⁻ tended to increase after Ar⁺ sputtering for 15 s; in addition, elemental K was gradually dominant with increasing sputter time to 90 s. It can be concluded that PDMS exists throughout the SEI, while there is a denser PDMS layer near K metal surface (Fig. 5c)[46]. To investigate the mechanical properties of the SEI on the K anode, atomic force microscopy (AFM) was conducted. The SEI formed in the PDMS-free

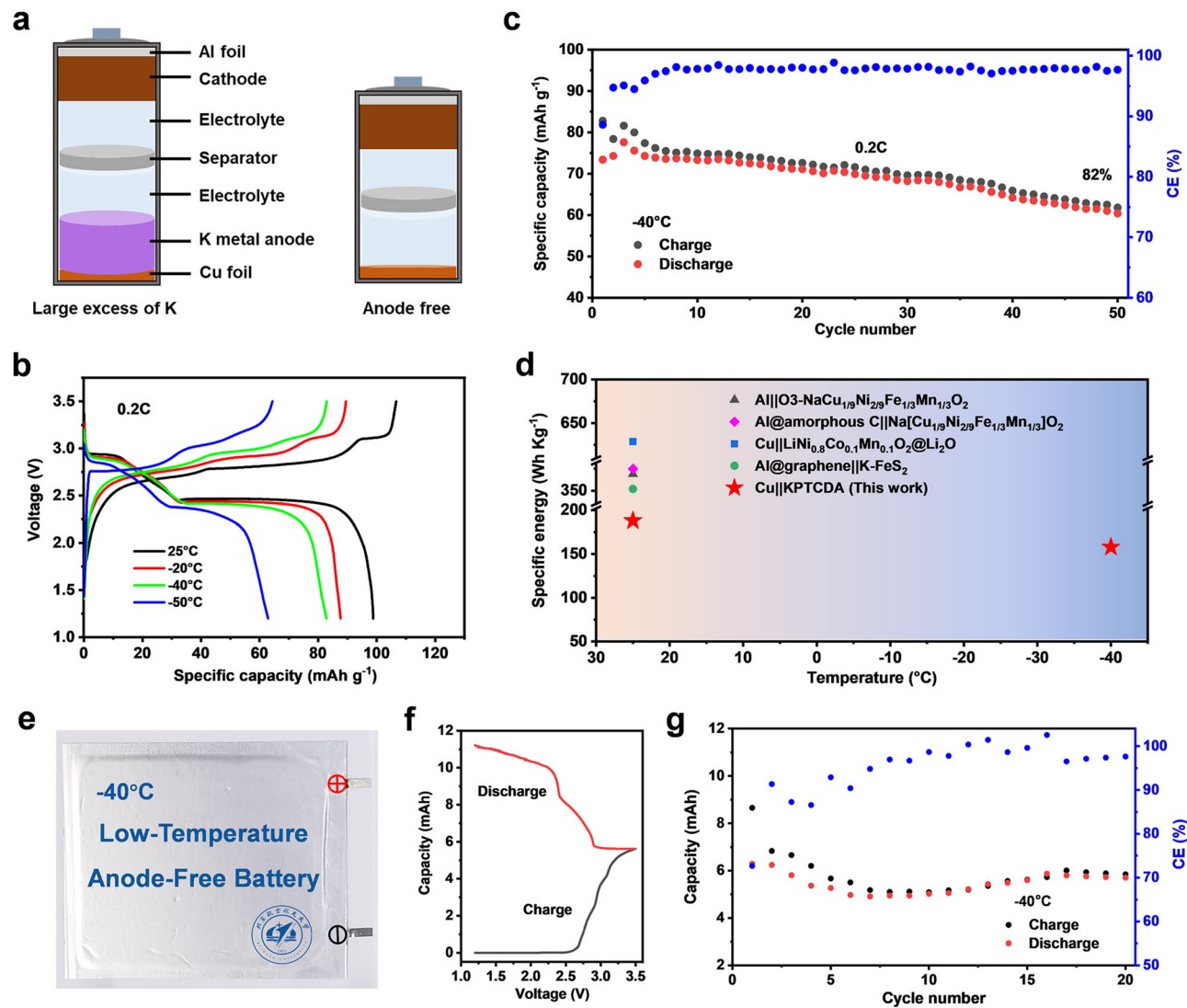

**Fig. 6 | The electrochemical performance of the anode-free Cu‖KPTCDA full batteries. a** Schematic illustration of the conventional K metal battery and anode-free battery configurations. **b** Voltage profiles of Cu‖KPTCDA coin cells at different temperatures. **c** Cycling performance of the Cu‖KPTCDA coin cells at 0.2 C (1 C = 130 mA g⁻¹) and −40 °C. The mass loading of the positive electrode is ~6.5 mg cm⁻². **d** Summarization of the representative anode-free metal coin cells by work temperature. The specific energy was calculated based on the total loading mass of the negative and positive electrodes. **e** Photograph of a pouch cell with a positive electrode mass loading of ~4.2 mg cm⁻². **f** Typical charge/discharge profiles and **g** cycling performance at 10 mA g⁻¹ and −40 °C. The specific capacities are calculated based on the mass of KPTCDA.

electrolyte possesses a Derjaguin–Müller–Toporov (DMT) modulus of ~1156 MPa (Supplementary Fig. 23a). In contrast, Si-O-Si linkages endow the SEI with outstanding mechanical strength, resulting in a significant increase in the DMT modulus up to 1489 MPa, which is mechanically strong to buffer the volume variations and suppress K dendrite growth during cycling (Supplementary Fig. 23b, c)[47]. The mechanically robust and ion-conductive SEI consisting of KF-rich phases and PDMS-derived moieties can guarantee highly reversible K plating/stripping, effectively improving the sustainability of the K anode (Fig. 5d, e)[9,39].

**Electrochemical performances of low-temperature anode-free Cu‖KPTCDA cells**

Encouraged by the highly reversible plating/stripping behaviour of K, low-temperature anode-free K metal full cells were assembled with a potassiated PTCDA (KPTCDA) positive electrode (Supplementary Figs. 24 and 25)[48]. In conventional K metal batteries, a large excess of K anodes is used, which seriously compromises the energy density at the cell level. For the anode-free cell architecture[49], the overuse of K can be avoided, which promises to improve the energy density (Fig. 6a). As a

proof, the anode-free Cu‖KPTCDA coin-type cell was constructed with the KDP electrolyte. The cyclic voltammetry (CV) curves exhibit three oxidation/reduction peaks at approximately 3.21/2.25, 2.99/2.58, and 2.79/2.84 V, accompanied by a slight increase in the polarization as the temperature decreases from 25 to −50 °C (Supplementary Fig. 26)[50]. Moreover, the Cu‖KPTCDA cell with a specific current of 0.2 C (1 C = 130 mA g⁻¹) shows a specific capacity of 98.6 mAh g⁻¹ at 25 °C. When the temperature is decreased to −40 °C, a reversible capacity of 82.8 mAh g⁻¹ is achieved with a high retention of 84%. Even at −50 °C, it still maintains 64% of the initial capacity at 25 °C, demonstrating the considerably advanced low-operating temperature performance (Fig. 6b). Then, the long-cycle performance of Cu‖KPTCDA cells at −40 °C was studied, and an 82% capacity retention of the initial capacity is attained at 0.2 C after 50 cycles (Fig. 6c). The corresponding voltage profiles display a discharge plateau at approximately 2.38 V, and the charge/discharge curves maintain a similar shape at different cycles (Supplementary Fig. 27). To the best of our knowledge, this is the first time that the working temperature range of anode-free cells has been extended to −40 °C, and the specific energy of our Cu‖

KPTCDA (152 Wh kg$^{-1}$ based on the total mass of the negative and positive electrodes without current collectors) is highly competitive among the previously reported Li/Na-ion full cells cycled at −40 °C (Fig. 6d, Supplementary Figs. 28 and 29, and Supplementary Tables 6 and 7)[2,39,51–55]. Owing to the easy assembly procedure of anode-free cells compared with conventional alkali metal batteries[2,39], a Cu‖KPTCDA anode-free pouch cell (5 cm × 7 cm) was successfully constructed (Fig. 6e). It delivers a high initial discharge capacity of 6.28 mAh with 90% capacity retention after 20 cycles at 10 mA g$^{-1}$ and −40 °C (Fig. 6f, g). Therefore, our work demonstrates the feasibility for fabricating low-temperature high-energy batteries by anode-free strategies.

## Discussion

In summary, we realized a low-temperature anode-free K metal battery through electrolyte regulation. The low-concentration KPF$_6$/DME with high ionic conductivity and weak solvation effect exhibits fast K$^+$ ion diffusion dynamics at low temperatures. Moreover, an integrated and homogeneous SEI with increased fractions of KF and Si−O-based components is formed via adding a graftable PDMS electrolyte additive, which guarantees a stable and highly reversible K plating/stripping behaviour. As a result, an average 99.80% CE of K‖Cu cells is achieved at −40 °C. Meanwhile, K‖K cells show desirable electrochemical performance at low temperature, such as reduced K deposition overpotential, excellent rate capability and prolonged cycle life. Accordingly, anode-free Cu‖KPTCDA full cells display a high capacity retention of 82% after 50 cycles at 0.2 C with a specific energy of 152 Wh kg$^{-1}$ based on the total mass of the negative and positive electrodes at −40 °C. These results would brighten the prospect of low-temperature high-energy-density rechargeable batteries.

## Methods

### Preparation of electrolytes

Four electrolytes were purchased from Dodo Chem: 0.8 M KPF$_6$-EC/DEC, 1 M KPF$_6$-DME, 1 M KFSI-EC/DEC and 3 M KFSI-DME. In addition, 1 M KPF$_6$-DME was diluted with 1,2-dimethoxyethane (DME, Sigma–Aldrich) solvent to prepare KPF$_6$/DME-based electrolytes with different concentrations. Note that DME was pretreated with fresh potassium foil (K, Aladdin) to eliminate traces of water before use. Polydimethylsiloxane (trimethylsiloxy terminated, M.W. 770, Alfa) was used as an electrolyte additive to obtain 0.4 M KPF$_6$-DME with 2 vol.% PDMS. All electrolytes were prepared in an Ar-filled glove box (O$_2$ < 0.01 ppm, H$_2$O < 0.01 ppm).

### Preparation of the PTCDA positive electrodes

First, perylene-3,4,9,10-tetracarboxylic dianhydride (PTCDA, Alfa) was pretreated by the reported methods[48], in which PTCDA was annealed at 450 °C for 4 h under Ar with a heating rate of 5 °C min$^{-1}$. Then, the PTCDA positive electrodes were prepared by blending PTCDA with Super P and carboxymethyl cellulose (CMC, Acros) at a weight ratio of 8:1:1 by using deionized water as the solvent. After stirring for 2 h, the homogeneous slurry was cast onto aluminium (Al) foils and dried at 60 °C for 24 h.

### Electrochemical measurements

The ionic conductivity ($\sigma$) of the electrolytes at various temperatures was measured by EIS, in which two parallel platinum (Pt) plates were used as the working and reference electrodes. Then, it can be calculated by the following equation:

$$\sigma = \frac{L}{SR} \qquad (1)$$

where L represents the distance between the two Pt electrodes, S is the area of Pt symmetrically placed in the electrolyte and R is the impedance obtained by the Nyquist plot.

For the K‖Cu half-cell experiments, conventional 2032-coin cells were assembled in an Ar-filled glovebox. Each half-cell consisted of a Cu electrode (19 mm diameter) and a K foil counter electrode (0.8 × 0.8 cm, the thickness is 300–500 μm) with Celgard 2400 separator (19 mm diameter) and 60 μL K$^+$-based electrolytes. The average CE was obtained by the most common Aurbach method, as follows: K was predeposited on the Cu foil at 0.5 mA cm$^{-2}$ for 10 h, then the plating/stripping process was performed at 1 mA cm$^{-2}$/1 mAh cm$^{-2}$ for cycles, and finally fully stripped to 1 V at 0.5 mA cm$^{-2}$. The average CE was calculated by the following equation:

$$Average\,CE = \frac{nQ_c + Q_S}{nQ_c + Q_T} \qquad (2)$$

where $nQ_C$ is the cumulative cycling capacity, $Q_S$ is the fully stripped capacity and $Q_T$ is the predeposited capacity.

EIS was measured using Cu‖K cells in the frequency range of 1 MHz–0.01 Hz (three points per decade of frequency) with an amplitude of 10 mV by potentiostatic signal after holding open circuit voltage for 300 s at quasi-stationary potential and then EIS plots were fitted by ZView (1400 celltest system, Solartron). For the K‖K symmetric cell, K foil replaced Cu as the working electrode, while the other processes remained the same. Prior to assembling anode-free K metal full cells, the PTCDA positive electrode was prepotassiated in half-cells at 0.2 C and 25 °C. Then, anode-free full cells were assembled by pairing the potassiated PTCDA positive electrode with Cu current collectors, and mass loading of the positive electrode is ~6.5 mg cm$^{-2}$. With the same prepotassiation process, anode-free pouch cells were assembled by using a 5 cm × 7 cm prepotassiated PTCDA positive electrode with a mass loading of ~4.2 mg cm$^{-2}$. Both Celgard 2400 and glass fibre were used as the hybrid separators, and the amount of electrolyte was controlled to be 100 μL for coin cells and 3 mL for pouch cells. The operating voltage of Cu‖KPTCDA full cells is 1.2-3.5 V. The CR2032-type coin cells and pouch cells were assembled in an Ar-filled glove box, and low-temperature electrochemical tests were carried out on CT 3001 A Land battery testing systems (Jinnuo Electronics Co. Ltd., China) in a cryogenic box (JK-80G, Kingjo). The specific energy was calculated by the following equation:

$$E_{cathode + anode} = \frac{U_{avg} \times Q_{cell}}{m_{KPTCDA} + m_{SuperP} + m_{CMC}} \qquad (3)$$

where $U_{avg}$ is the average cell operating voltage; $Q_{Cell}$ is the total capacity of the cell; ($m_{KPTCDA} + m_{CMC} + m_{Super\,P}$) is the total loading mass of the positive electrode after prepotassiation; the weight of the negative electrode is 0; current collectors is not considered in the formula.

### Characterization

The freezing points of 0.4 M KPF$_6$-DME electrolytes was tested by differential scanning calorimetry (DSC, NETZSCH DSC 214, Germany) with a liquid nitrogen cooling system. In situ temperature-dependent Raman spectra of electrolytes with different KPF$_6$ concentrations were collected by a LabRAM HR Evolution with an excitation wavelength of 633 nm from 25 °C to −50 °C. Attenuated total reflection-Fourier transform infrared (ATR-FTIR) spectra were measured by a Bruker TENSOR II, and samples were protected by an argon-filled bag during transfer, and sample's surface was in close contact with ATR mode detectors, thus avoiding air exposure during testing process. The morphologies of K were observed by field emission scanning electron microscopy (JEOL-7500), and the samples were sealed in an argon-filled bag and then quickly loaded on the SEM holder. X-ray photoelectron spectroscopy (XPS) was performed by an ESCALAB 250Xi, and the spectra of the SEI were analysed based on the C 1s

binding energy of 284.6 eV. A semi-in situ vacuum transfer device under was used to transfer samples before XPS spectra were collected. The Raman mapping of cycled K metal electrodes was obtained by a SENTERRA II (Bruker) with a 532 nm excitation wavelength. Note that dissembled K electrodes were rinsed with DME solvent and sealed in a hermetic device for further characterizations. TOF-SIMS was conducted with a PHI nano TOF II (ULVAC-PHI Inc., Japan). The K metal sample was directly transferred from the glove box to the TOF-SIMS vacuum chamber by a special transfer vessel without being exposed to air. Sputter etching was conducted using an $Ar^+$ beam (3 kV 100 nA) to obtain the desired depth profile. The area of analysis was $50 \times 50\ \mu m^2$. In the AFM (Bruker Corp., Dimension Icon) experiment, the $K$ anode was scanned by a silicon AFM tip (Bruker Corp., $k = 26\ N\ m^{-1}$, $f_0 = 300\ kHz$) to acquire the DMT modulus in the Ar-filled glovebox.

## Calculation methods

All first-principles calculations were performed in the Gaussian 09 D.01 program package using the Lee-Yang-Parr correlation functional (B3LYP) combined with the 6-311G* basis set. Visual Molecular Dynamics (VMD) software[56] was used to present the molecular structures. Solvent effects were considered by the implicit solvation model based on density (SMD) (DiethylEther). The deslovation energy ($E_{desolvation}$) was calculated with the equation below:

$$E_{desolvation} = E_{K^+-solvent/PF_6^-} - x \times E_{solvent/PF_6^-} - E_{K^+} \qquad (4)$$

The MD calculations were performed by the GROMACS 5.3 package[57]. The force field was chosen with OPLS-AA[58]. The electrolyte systems were equilibrated for 1000 ps (time step of 2 fs) followed by a 10 ps (time step of 2 fs) simulation run in an isothermal-isobaric ensemble (NPT). The temperature was controlled to 233 K by a V-rescale thermostat. VMD was used to sample snapshots of the solvation shells and analyse the $K^+$ RDF of different electrolytes from the simulation trajectory.

To analyse the deposition behaviour of $K^+$, first-principles calculation on the basis of DFT were conducted via the Vienna Ab initio Simulation Package (VASP). The Perdew–Burke–Ernzerhof (PBE) functional under the generalized gradient approximation (GGA) was employed to describe the exchange–correlation. The projector augmented wave (PAW) pseudopotentials were adopted, and the cut-off energy of plane-wave basis was set to 400 eV. The convergence criteria of total energy and force were set to be $10^{-4}$ eV and 0.03 eV/Å, respectively. A $1 \times 1 \times 1$ Monkhorst–Pack $K$-point grid was used considering the large-scale systems. VdW-DF2 functional was used to describe van der Waals (vdW) force physical interaction. The differences of charge density were conducted by using VESTA software package. To quantitatively describe the binding ability of loading materials, the binding energy is defined as below:

$$E_{binding} = E_{K^+-surf} - E_{surf} - E_{K^+} \qquad (5)$$

where $E_{surf}$, $E_{K^+}$ and $E_{K^+-surf}$ are total energies of K (001) or PDMS-K (001) surface, $K^+$, and the adsorption system, respectively.

## Data availability

The data supporting the findings of this study are available from the authors upon request.

## Code availability

The codes that support the findings of this study are available from the corresponding author upon request.

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

## Acknowledgements

The authors acknowledge the financial support of the National Natural Science Foundation (NSFC) of China (21972007 and 52172178 to H.W., U1910208 to L.G.), the Natural Science Foundation of Beijing (2222059 to H.W.), and International Cooperation Project of National Key Research and Development Program of China (2022YFE0126300 to H.W.). The authors thank Professor Tinglu Song from Beijing Institute of Technology for the technical support in TOF-SIMS. The authors also thank Professor Rui Wen from Chinese Academy of Sciences for the technical support in AFM. This research was supported by the high performance computing (HPC) resource at Beihang University.

## Author contributions

H.W. and L.G. conceived the idea and designed this work; M.T. and S.D. carried out the electrolyte regulation and electrochemical measurements; Q.Z. performed the DFT calculation. M.T., S.D., and L.C. carried out the ex situ or in situ characterizations and analysis; M.T., J.W., Y.L., X.Y., and H.W. wrote the manuscript and participated in the discussion to improve the paper.

## Competing interests

The authors declare no competing interests.

 
