## [Peer Review File · Nature Communications]

REVIEWER COMMENTS

Reviewer #1 (Remarks to the Author):

The authors report on the impressive electrochemical performance of anode free K-metal battery operating at -40C. The work is quite consistent and the reported claims are well ground on the included results and discussion. I believe the work is of great interest for a wide readership and urgent enough to be published in Nature Communications. (apologies for my delay). My main concern which remains unanswered is that as Fig S9 shows there is a strong influence of the temperature on the overpotential and while this batter shows impressive results at -40C, the results are not that interesting at room temperature....so I wonder.... First... will this battery be useful at all... and how could this be explained?? (but you can probably discuss this in a different manuscript) and also.... Is there a different optimum amount of additive at room temperature (fig 1d suggests so)? As said, leaving these questions open I feel the work is of high enough quality to make it worth publishing in Nature Communications.

Besides this I have some minor comments:

I can't see any difference between Figure 1g and Figure S4b. The authors have probably made a mistake and used the same plot twice? It is therefore difficult to follow the corresponding discussion.

p.3, line 71, the authors express per kg of cathode+anode... however I guess it is based only on weight of active material...so no current collector.. if so...are you considering the weight of the cathode after prepotassiation?

p.6, lines 116 and 117. It is not clear what is meant...do you men that it is formed (as suggested in the ATR)? Or that it could just form?

Figure S12. What is the top spectrum vs the bottom spectrum?

Figure 3d. units are missing on the y-axis.

p.10, lines 183-185 needs rewriting.

Figure 5 d suggests PDMS layer is several microns thick, but the models were based on PDMS monolayer or is it assumed that the electrolyte travels through the deposited PDMS... a clarification or discussion is needed.

p.13, line 222. The authors claim the energy density for the full cell is even larger than that reported for full Li or Na cells at -40C, but actually in Figure 6d there is no comparison to results of Li or Na full cells at 40C... either add comparative data or remove the statement

The authors report O₂ and H₂O levels below 0.01ppm. Is this real?? Or should it read 0.1ppm? Please provide details of the glovebox brand and sensors.

Why is the PTCDA annealed at 450C?

Are the potassium or Cu foils square? It says 0.8 x 0.8 cm. What cell type was used? Was it coin cell?

A special transfer vessel is used to transfer rinsed electrodes to the TFO-SIMS chamber... what about other characterization techniques such as SEM, IR, or XPS?

Reviewer #2 (Remarks to the Author):

Overall this is an interesting study which shows how electrolyte additives can give useable potassium ion batteries at low temperature. In terms of the specific application (low temperature potassium ion, this is quite niche and there is insufficient comparison of the results to room temperature (or range of temperatures) potassium ion such that the wider applicability of the work can be inferred. Alternatively, a comparison across lithium ion and sodium ion at -40 also would be significant. I feel the work needs to expand into either of these directions to meet the significance required for Nature communications. The following are more specific comments

1. In introduction, the authors could add specific applications where low temperature K-anode free batteries would be useful?

2. The DSC data explained (given in Fig 3a-SI) should be explained properly to highlight the results and significance of performing DSC.

3. The EIS data presented in fig 5 (SI) is an important test to claim high ionic conductivity of 0.4 M electrolyte at -40 °C. However, the current analysis lacks proper evidence. The following data must be provided to support the above claim:

3a. The EIS data before cycling and after 1st cycle should also be provided to help understand the cell resistance evolution over the initial cycles.

3b. Please properly fit the EIS data provided in Fig 5 (SI), calculating the contribution from each resistance component.

3c. The authors claim high ionic conductivity for 0.4 M electrolyte, therefore the K⁺ diffusivity should be calculated by using Warburg component of the Nyquist plot.

4. The authors claim (line 110-111) that the cloudy electrolyte (0.4 M NaPF₆ – DME) was due to decomposition of DME solvent. Can the authors provide some references for that, or could this cloudy behaviour be due to the re-precipitation of KPF₆ salt over time?

If not, please provide evidence of DME decomposition with some chemical characterization of the electrolyte (without PDMS additive) after 48 h.

5. Does the addition of PDMS change the ionic conductivity of electrolyte at ambient or low temperatures as compared to no PDMS additive (data given in Fig 3-SI is without PDMS addition) ? What is the freezing point of PDMS and can it function at -40 °C ?

6. Reason/evidence should be provided as to why 2 vol. % is “optimal” quantity of PDMS as compared to 4 and 8 vol. % (Fig 8-SI)?

7. It's interesting to see low overpotentials with 0.4 M KDP at -40 C (fig 3f). However, can the authors compare overpotentials at 25 °C and -40 °C and explain the reason for low overpotentials obtained at -40 °C in K/K symmetric cells?

8. EIS data in Fig 4 c should be properly fitted to show the evolution of resistance due to SEI layer ?

9. Do the SEI components form at $-40\text{ }^{\circ}\text{C}$ temperature dependent? Would the same SEI components form even at room temperature? The authors should provide relative ratios of SEI components at ambient temperature in comparison to $-40\text{ }^{\circ}\text{C}$.

10. How did the authors reach to the conclusion that the SEI without PDMS would have cracks and covered with organics as the top layer (red)? References should be provided to support such claim.

11. Did the cell fail after just 3 cycles when 1 M KPF6 (DME) was used with Cu/KPTCDA full battery ? A comparative voltage – specific capacity profile of all 3 electrolytes should be given in fig 15 – SI. Also please revise the caption since it is vague whether the data (Fig 15b – SI) given is for K/Cu half cell or Cu/KPTCDA full cell.

12. The authors should provide details (formulae and table of values) as how they reached/calculated the energy density values of 188 Wh Kg^{-1} and 152 Wh Kg^{-1} . Currently, it is unknown what parameters (voltage etc.) were incorporated to calculate these values.

13. Are the energy density values calculated on coin cell or pouch cell level ? The energy density values at pouch cell level should also be provided to determine potential this system holds as claimed by the authors ?

14. How these energy density values compared with anode free Li and Na based batteries (at low temperatures) to validate the need to develop K anode free batteries.

15. The authors have shown low cycle no. (50 cycles in coin cell) for K anode-free battery. What do the authors think about this low cycle no. performance?

16. The electrolyte used is a dilute KPF6 would run out quickly (as compared to high molar ratios) during cycling of anode-free batteries. Therefore, what is the significance of using low conc. electrolyte in this work, irrespective of its functionality at low temperatures?

17. Overall language of the manuscript could be improved for better understanding. The current version lacks proper sentence structure.

Reviewer #3 (Remarks to the Author):

It is an interesting study on low temperature behaviors of potassium (K) battery chemistry. Nevertheless, the significance of the study needs improvement as the temperature is mostly limited at -40°C (A phase transition, which might be due to salt precipitation, was observed in differential scanning calorimetry, indicating the temperature limitation of formulated electrolyte is around -50°C). The state of the art studies demonstrates low temperature performance of alkali metal batteries down to -85°C, such as DOI: 10.1002/anie.201900266 and doi.org/10.1038/s41560-019-0474-3. Additionally, some mechanisms are not well explained and some critical experimental/analytical details are missing in the current manuscript.

1. The average coulomb efficiency (CE) of KPF6-DME system is lower than that of KFSI-EC/DEC in supplementary Fig.1. Why it is selected for low temperature study? What is more, all the samples show irregular morphology of deposition on current collector (supplementary Fig. 2). What is the logic to choose KPF6-DME over others? Besides, the screening was performed at room temperature, which is not relevant to low temperature investigation. Low temperature screening is required.

2. In supplementary Fig. 5, EIS fitting and equivalent circuit are required to determine the charge-transfer resistance. Why the EIS tests were conducted after 5 cycles? What are the EIS profiles at 1st cycle?

3. The authors claimed that DME can react with K metal (line 108 and 109 on page 6). Then why the K metal was used to pretreat DME solvent (line 253 on page 15)?

4. In supplementary Fig. 8, the authors stated that 2 vol % is the optimized additive concentration. However, no values less than 2 vol % were investigated. The authors are required to provide supporting results.

5. In Fig. 3a, the authors need to reveal the initial 20 cycles of CE in the figure inset. For Fig. 3e, the authors are required to provide the corresponding rate capability test as a plot of capacity vs cycle with CE. What is the recovery rate when current density goes back to 0.5 mA cm⁻²?

6. In supplementary Fig. 10, the surfaces of K metals using the formulated electrolyte are not uniform but porous. The author need to provide explanation.

7. In Fig. 4c, both equivalent circuit and electrochemical impedance spectroscopy (EIS) fitting are needed. The authors need to provide quantitative analysis.
8. The formation and protection mechanism of organic-inorganic hybrid interphase (line 187 on page 10) are unclear. How are the mechanical properties of the interphase? More relevant characterization tools (i.e. AFM) are recommended.
9. DSC profile (supplementary Fig. 3a) is not complete, both cooling and heating curves need to be revealed.
10. Temperature dependent ionic conductivity of formulated electrolyte (with additive) need to be presented and compared.
11. The additive amounts vary in K || Cu (K || K) cells and Cu || KPTCDA cells. For K || Cu and K || K cells, 60 μL electrolyte (1.2 μL additive) is added. While for Cu || KPTCDA cells, 100 μL electrolyte (2 μL additive) is used for coin cells and 3000 μL (60 μL additive) for pouch cells. What is the logic to use different electrolyte amounts?
12. In line 223 on page 13, the authors discussed that the cell shows high safety. Nonetheless, no supporting results are provided. The authors need to provide supportive results (i.e. DSC analysis).
13. For Fig. 6g, the authors need to provide both charge capacity and CE. What is more, temperature dependent cyclic voltammetry and galvanostatic charge-discharge profile are need for Cu || KPTCDA cells in Fig. 6.

Response to Reviewers for “Low-Temperature Anode-Free Potassium Metal Batteries”

Reviewer #1 (Remarks to the Author):

The authors report on the impressive electrochemical performance of anode free K-metal battery operating at -40°C . The work is quite consistent and the reported claims are well ground on the included results and discussion. I believe the work is of great interest for a wide readership and urgent enough to be published in Nature Communications. (apologies for my delay). My main concern which remains unanswered is that as Fig S9 shows there is a strong influence of the temperature on the overpotential and while this batter shows impressive results at -40°C , the results are not that interesting at room temperature....so I wonder.... First... will this battery be useful at all... and how could this be explained?? (but you can probably discuss this in a different manuscript) and also.... Is there a different optimum amount of additive at room temperature (fig 1d suggests so)? As said, leaving these questions open I feel the work is of high enough quality to make it worth publishing in Nature Communications.

Reply: Thanks very much for the reviewer’s positive comment on our work.

Regarding the issues of temperature-dependent overpotential, we have redrawn the graph to clearly illustrate this issue. As shown in **Supplementary Figure 17d**, the overpotentials of K||K cells obtained at -40°C is only slightly higher than that of K||K cells at 25°C at current densities from 0.5 to 4 mA cm^{-2} , indicating that the overpotentials of the K||K cells in the $0.4\text{ M KPF}_6\text{-DME}$ with 2 vol. \% polydimethylsiloxane (PDMS) additive (referred to as the KDP electrolyte) is weakly influenced by the temperature. This result can be ascribed to the stable solvation configuration and high ionic conductivity of the 0.4 M electrolytes over a wide temperature range (**Fig. 1**). Moreover, the rate performance at 25°C is also much better than that reported for potassium metal anodes, i.e. high current density and small overpotential (**Figure R1**). Thus, we also have achieved good electrochemical performance of K metal anodes with the KDP electrolyte at room temperature.

According to the reviewer’s suggestion, we have attempted to change the amount of the PDMS additive for a better CE at 25°C . With the PDMS contents increasing from 1 to 8% , the CE values of K||Cu cells are 98.56% , 99.23% , 99.35% , 99.34% , 99.17% , 99.16% and 98.64% , respectively, and the CE of 3% and 4% PDMS was only slightly improved compared to the 2% PDMS used in this work (**Figure R2**). Nevertheless, it should be noted that this CE value obtained at 25°C is still not as good as the CE obtained at -40°C , which can be ascribed to the enhanced reactivity of DME with K as the temperature rising. To solve this issue, we find another effective additive that can significantly improve the average CE up to 99.7% at 25°C , and it will be reported in the future work (**Supplementary Figure 28b**).

Overall, we achieve an excellent rate capability of K metal anode at room temperature. More importantly, for the first time, a highly reversible K plating/stripping behavior is achieved at low temperature. Encouraged by this, we successfully extend the working temperature range of anode-free cells down to -40°C . Our work will spur a new wave of research towards low-temperature and high-energy-density batteries by anode-free strategies.

Supplementary Figure 17. The rate performance of K||K cells with the KDP electrolyte at (a) 25°C and (b) -40°C, respectively; (c) Representative plating/stripping curves at different current densities at 25°C; (d) Comparison of overpotentials obtained from (a) and (b).

Figure R1. The comparisons of rate capability with reported K metal anodes at 25°C.

Figure R2. The CE tests of K||Cu cells at 1 mA cm⁻² and 25°C.

Supplementary Figure 28b. The CE tests of K||Cu half cells with various electrolytes at 1.0 mA cm⁻²/1 mAh cm⁻².

Besides this I have some minor comments:

1. I can't see any difference between Figure 1g and Figure S4b. The authors have probably made a mistake and used the same plot twice? It is therefore difficult to follow the corresponding discussion.

Reply: Thanks for the reviewer's question. We have carefully checked these two figures, and found that they are two different plots, which describe the RDF data at 233K for the 0.4 M and 1 M KPF₆-DME electrolytes, respectively. To clearly illustrate the difference between **Figure 1g** and **Figure S4b**, we put the correlation plots together in **Figure R3**. Combined with the calculated coordination number shown in **Supplementary Table 2**, it can be found that PF₆⁻ anions in 1 M KPF₆-DME has a higher possibility of entering the first K⁺ solvation sheath than that in 0.4 M KPF₆-DME, but in reverse for DME.

Figure R3. RDF data of (a) 0.4 M and (b) 1 M KPF₆-DME electrolytes at 233K.

Supplementary Table 2. The calculated coordination number in 0.4 M and 1 M KPF₆-DME at 233K.

	Coordination number at 233K	
	0.4 M KPF ₆ -DME	1 M KPF ₆ -DME
K-O (at 4.1 Å)	6.787	6.048
K-P (at 5.1 Å)	0.598	0.938
K-F (at 3.8 Å)	1.365	2.129

2. p.3, line 71, the authors express per kg of cathode+anode... however I guess it is based only on weight of active material...so no current collector. if so...are you considering the weight of the cathode after prepotassiation?

Reply: We thank the reviewer for pointing out this. The specific energy density of Cu||KPTCDA cells is calculated based on the total loading mass of cathode and anode instead of the current collectors. The cathode includes the prepotassiated PTCDA, CMC and Super P, and the anode is a bare Cu current collector.

In the revised manuscript on page 20, line 331-334, we added:

$$E_{\text{cathode+anode}} = \frac{U_{\text{avg}} \times Q_{\text{cell}}}{m_{\text{KPTCDA}} + m_{\text{CMC}} + m_{\text{Super P}}}$$

where U_{avg} is the average cell operating voltage; Q_{cell} is the discharge capacity of the cell; $(m_{\text{KPTCDA}} + m_{\text{CMC}} + m_{\text{Super P}})$ is the total loading mass of cathode after prepotassiation; The weight of anode is 0; Current collectors is not included in the formula.

3. p.6, lines 116 and 117. It is not clear what is meant...do you mean that it is formed (as suggested in the ATR)? Or that it could just form?

Reply: Yes, it is formed. To illustrate this part clearly, we have rewritten the related description of the evolution process of PDMS.

Revised text on page 6, line 118-120:

Besides, Si-CH₃ (1261 cm⁻¹) reacts to form Si-OH (3496 cm⁻¹), and PDMS could be crosslinked with neighbors by Si-O-Si bonds to form polysiloxane, which is facilitated by an electrical field (Supplementary Fig. 9)²⁷.

4. Figure S12. What is the top spectrum vs the bottom spectrum?

Reply: Thanks for the reviewer's question. The top spectrum represented the Si 2p signal in the outer SEI on the K surface, and the bottom spectrum represented the Si 2p signal of the inner SEI after Ar⁺ sputtering for 100 s.

In the reversed manuscript, we changed:

Supplementary Figure 22b. In-depth Si 2p XPS spectra of the cycled K anode in the KDP electrolyte at -40°C .

5. Figure 3d. units are missing on the y-axis.

Reply: Thanks very much for the reviewer’s careful review. We have added the units on the y-axis of Figure 3d.

Figure 3. (e) Rate capability of the K||K cells in the KDP electrolyte at 25 and -40°C , respectively.

6. p.10, lines 183-185 needs rewriting.

Reply: We thank the reviewer’s suggestion. We have rewritten this part.

Revised text on page 12, line 206-208:

The Raman signal-mapping images of Si-O-Si (PDMS) and KF shown in Fig. 4i are generated from their peak intensities, enabling the clear visualization of the spatial distribution of KF and PDMS on the K surface.

7. Figure 5d suggests PDMS layer is several microns thick, but the models were based on PDMS monolayer or is it assumed that the electrolyte travels through the deposited PDMS... a clarification or discussion is needed.

Reply: Thanks for the reviewer's insightful review. In fact, we cannot make any conclusions about the thickness of the PDMS layer in the current version. Based on the test results of TOF-SIMS (Fig. 5) and in-depth XPS (Fig. 4), it can be found that PDMS components exist throughout the SEI, while there is a denser PDMS layer near the K metal surface, and its thickness may be tens of nanometer. To study the bonding mode between the inner PDMS layer and K, a simplified models based on PDMS monolayer are established. Meanwhile, it is reasonable to assume that the PDMS layer, as a part of the SEI formed on the K metal surface, facilitates the K-ion flux and homogeneous K electrodeposition, in consideration of the enhanced electrochemical performance of K||Cu and K||K cells. Thus, the models based on PDMS monolayer is just a schematic to illustrate that the PDMS is grafted on the K metal surface and act as a protective layer, which does not affect the correctness of the experimental results.

Revised text on page 14, line 226-227:

It can be concluded that PDMS exists throughout the SEI, while there is a denser PDMS layer near the K metal surface (Fig. 5c)⁴⁹.

8. p.13, line 222. The authors claim the energy density for the full cell is even larger than that reported for full Li or Na cells at -40°C, but actually in Figure 6d there is no comparison to results of Li or Na full cells at -40°C... either add comparative data or remove the statement

Reply: Thanks for the reviewer's good suggestion. Given the scope of this work, we focused on the anode-free cells for a comparison of energy densities in Fig. 6d. Furthermore, the energy density (calculated based on the total loading mass of cathode and anode) of our Cu||KPTCDA cells is highly comparable to Li/Na-ion full cells at -40°C, and the related comparative data have been added in Supplementary Figure 29, Table 7.

Revised text:

Supplementary Figure 29. Comparison of specific energy with representative Li/Na-ion full cells at -40°C. The specific energy densities were calculated based on the total loading mass of

cathode and anode. Note: the data listed in **Supplementary Table 7** is used for the plot.

Supplementary Table 7. Comparisons of the electrochemical performance between this work and the reported low-temperature rechargeable batteries.

Cell Configuration	Operating Temperature (°C)	Current rate (A g ⁻¹)	Average discharge voltage(V)	Specific energy (Wh kg ⁻¹)	Ref.
Cu KPTCDA	-40	0.026	2.39	152	This work
Bi Na ₄ Fe ₃ (PO ₄) ₂ P ₂ O ₇	-40	0.02	2.3	127.5	Angew. Chem. Int. Ed. 61 , e202116930 (2022)
HC NFPF@C@MCNTs	-40	0.037	2.1	39.4	Small 18 , 2204830 (2022)
HC P2-NaMNNb	-40	0.037	3.2	188	Nat. Commun. 13 , 3205 (2022)
AG PTPAn	-40	0.01	2.5	118	Angew. Chem. Int. Ed. 60 , 23858–23862 (2021)
NC LiNi _{0.65} Co _{0.15} Mn _{0.2} O ₂	-40	0.012	2.44	146.8	Angew. Chem. Int. Ed. 61 , e202209619 (2022)
Gr NMC811	-40	0.02	3.4	129	Angew. Chem. Int. Ed. 134 , e202205967 (2022)
Li _x C Li ₂ V ₂ (PO ₄) ₃	-40	0.017	3.92	139	Angew. Chem. Int. Ed. 56 , 16606–16610 (2017)

Note: the energy density is calculated based on the total loading mass of cathode and anode.

9. The authors report O₂ and H₂O levels below 0.01ppm. Is this real?? Or should it read 0.1ppm? Please provide details of the glovebox brand and sensors.

Reply: Yes, O₂ and H₂O levels are both below 0.01ppm as shown in **Figure R4**. The glovebox (Mikrouna Super 1220/750/900, Shanghai Mikrouna Mech. Tech. Co., Limited), O₂ probe (MK-OX-SEN1, VTI) and H₂O probe (MK-XTR-100, VTI) are used in this manuscript.

Figure R4. Operation interface of the glovebox.

10. Why is the PTCDA annealed at 450°C?

Reply: Thanks for the reviewer's good question. Annealing PTCDA at 450°C can increase its electric conductivity while minimizing its thermal decomposition, and this phenomenon has been widely demonstrated by previous literatures (*Adv. Mater.* **30**, e1805486 (2018); *Energy Storage Mater.* **2**, 63-68 (2016)).

11. Are the potassium or Cu foils square? It says 0.8 x 0.8 cm. What cell type was used? Was it coin cell?

Reply: Thanks for the reviewer's question. K foil is square (0.8×0.8 cm), and Cu foil is circular with a diameter of 19 mm. 2032-coin cells were used for K||Cu cells. We have added these experimental details in the revised manuscript.

Revised text on page 19, line 308-310:

For the K||Cu half-cell experiments, conventional 2032-coin cells were assembled in an Ar-filled glovebox. Each half-cell consisted of a round Cu electrode (19 mm diameter) and a square K foil counter electrode (0.8×0.8 cm) with Celgard 2400 separator and 60 μL K⁺-based electrolytes.

12. A special transfer vessel is used to transfer rinsed electrodes to the TOF-SIMS chamber... what about other characterization techniques such as SEM, IR, or XPS?

Reply: Thanks for the reviewer's question. As for the XPS tests, a semi-in situ vacuum transfer device was used to transfer samples before collecting XPS spectra. For SEM tests, samples were sealed in an argon-filled bag, and quickly loaded on the SEM holder. For the IR tests, samples were protected by an argon-filled bag during transfer, and sample's surface was in close contact with ATR mode detectors, thus avoiding air exposure during testing process. We have added these experimental details in the revised manuscript.

Revised text on page 21, line 340-346:

For XPS tests: A semi-in situ vacuum transfer device was used to transfer samples before collecting XPS spectra.

For SEM tests: Samples were sealed in an argon-filled bag, and quickly loaded on the SEM holder.

For IR tests: Samples were protected by an argon-filled bag during transfer, and sample's surface was in close contact with ATR mode detectors, thus avoiding air exposure during testing process.

Reviewer #2 (Remarks to the Author):

Overall this is an interesting study which shows how electrolyte additives can give useable

potassium ion batteries at low temperature. In terms of the specific application (low temperature potassium ion, this is quite niche and there is insufficient comparison of the results to room temperature (or range of temperatures) potassium ion such that the wider applicability of the work can be inferred. Alternatively, a comparison across lithium ion and sodium ion at -40°C also would be significant. I feel the work needs to expand into either of these directions to meet the significance required for Nature communications. The following are more specific comments

Reply: Thanks for the reviewer’s constructive suggestion on our work.

A comprehensive comparison of energy densities across lithium ion and sodium ion at -40°C has been discussed. Currently, low-temperature Li and Na batteries are divided into two categories: (1) Li/Na metal batteries. It has been reported that they always use a large excess of alkali metal anodes, which seriously compromises the specific energy density. Thus, our anode-free battery must have a higher energy density than theirs; (2) Li/Na ion full batteries (without Li/Na metal anodes). By comparison, the energy density (calculated based on the total loading mass of cathode and anode) of our Cu||KPTCDA full cell is highly comparable to that of Li/Na-ion full cells at -40°C (Supplementary Figure 29, Table 7).

Besides, to the best of our knowledge, there have been no report of low-temperature K full cells (excluding excessive use of K metal anodes) at -40°C . Overall, our Cu||KPTCDA full cells exhibit a top-level performance for the low-temperature K full batteries, and highly comparable among Li/Na-ion full cells at -40°C . The successful implementation of our work also provides an anode-free strategy to realize the low-temperature high-energy batteries. Thus, we hope that this version will meet the significance required for Nature communications.

Supplementary Figure 29. Comparison of specific energy with representative Li/Na-ion full cells at -40°C . The specific energy densities were calculated based on the total loading mass of cathode and anode. Note: the data listed in Supplementary Table 7 is used for the plot.

Supplementary Table 7. Comparisons of the electrochemical performance between this work and the reported low-temperature rechargeable batteries.

Cell Configuration	Operating Temperature (°C)	Current rate (A g ⁻¹)	Average discharge voltage(V)	Specific energy (Wh kg ⁻¹)	Ref.
----------------------------	-----------------------------------	------------------------------	--	------

Cu KPTCDA	-40	0.026	2.39	152	This work
Bi Na ₄ Fe ₃ (PO ₄) ₂ P ₂ O ₇	-40	0.02	2.3	127.5	Angew. Chem. Int. Ed. 61 , e202116930 (2022)
HC NFPF@C@MCNTs	-40	0.037	2.1	39.4	Small 18 , 2204830 (2022)
HC P2-NaMNNb	-40	0.037	3.2	188	Nat. Commun. 13 , 3205 (2022)
AG PTPAn	-40	0.01	2.5	118	Angew. Chem. Int. Ed. 60 , 23858–23862 (2021)
NC LiNi _{0.65} Co _{0.15} Mn _{0.2} O ₂	-40	0.012	2.44	146.8	Angew. Chem. Int. Ed. 61 , e202209619 (2022)
Gr NMC811	-40	0.02	3.4	129	Angew. Chem. Int. Ed. 134 , e202205967 (2022)
Li _x C Li ₂ V ₂ (PO ₄) ₃	-40	0.017	3.92	139	Angew. Chem. Int. Ed. 56 , 16606–16610 (2017)

Note: the energy density is calculated based on the total loading mass of cathode and anode.

1. In introduction, the authors could add specific applications where low temperature K-anode free batteries would be useful?

Reply: Thanks to the reviewer for the good suggestion. The potential practical applications have been added to the Introduction part.

Revised text on page 4, line 69-71:

Benefit from this, the anode-free Cu||prepotassiated 3,4,9,10-perylene-tetracarboxylic acid-dianhydride (KPTCDA) full cell is assembled and exhibit a high energy density of 152 Wh kg⁻¹_{cathode+anode} at -40°C. This result may spur the potential application of anode-free K metal batteries in powering unmanned aerial vehicles and rovers for the polar exploration.

2. The DSC data explained (given in Fig 3a-SI) should be explained properly to highlight the results and significance of performing DSC.

Reply: Thanks for the reviewer's good suggestion.

We conducted DSC tests to reveal the thermal stability of the KDP electrolyte. It can be found that an obviously exothermic peak occurred from the cooling curves. Based on this, phase transition temperature of our KDP electrolyte was identified at around -54°C (Supplementary Figure 15b). So, the following electrochemical tests were performed in the temperature range from 25 to -50°C.

Revised text on page 9, line 155-158:

Moreover, K||K cells with the KDP electrolyte can stably cycle over the temperature range of 25°C to -50°C, and the sudden increase of the overvoltage at -60°C is due to phase transition of the electrolyte, which is identified by the observable endothermic peak arising around -54°C through differential scanning calorimetry (DSC).

Supplementary Figure 15b. The DSC test of the KDP electrolyte.

3. The EIS data presented in fig 5 (SI) is an important test to claim high ionic conductivity of 0.4 M electrolyte at -40 °C. However, the current analysis lacks proper evidence. The following data must be provided to support the above claim:

3a. The EIS data before cycling and after 1st cycle should also be provided to help understand the cell resistance evolution over the initial cycles.

3b. Please properly fit the EIS data provided in Fig 5 (SI), calculating the contribution from each resistance component.

3c. The authors claim high ionic conductivity for 0.4 M electrolyte, therefore the K^+ diffusivity should be calculated by using Warburg component of the Nyquist plot.

Reply: We are very grateful for the reviewer's insightful review and constructive comments. To address the comments raised in parts a, b, and c, we have added additional EIS data of pristine and 1st cycle, the new figure section (equivalent circuit), table (fitting parameters), and the K^+ diffusivity to the manuscript in **Supplementary Figure 6**. After 1st cycle activation process, the impedance of K||Cu cells exhibits an obvious decrease in both 0.4 and 1 M KPF₆-DME at -40°C. Based on the fitted results of Nyquist plots at the 5th cycle, the 0.4 M KPF₆-DME exhibits a low R_{ct} of 759 Ω compared with 1 M KPF₆-DME (R_{ct} =1638 Ω), demonstrating the decreased energy barrier for K^+ ion desolvation process. Furthermore, the calculated D_{K^+} of 0.4 M KPF₆-DME is $1.37 \times 10^{-11} \text{ cm}^2 \text{ s}^{-1}$, higher than that of 1 M ($2.03 \times 10^{-12} \text{ cm}^2 \text{ s}^{-1}$) at -40°C. Thus, the 0.4 M KPF₆-DME exhibits high ionic conductivity and fast charge transfer kinetics at low temperature.

In the reversed manuscript, we added:

Supplementary Figure 6. EIS tests of K||Cu cells in (a) 0.4 M and (b) 1 M KPF₆-DME, respectively, at -40°C; (c) Fitted Nyquist plots at 5th cycle; (d) Equivalent circuit model and the corresponding impedance data; (e) The plot of Z'-ω^{-1/2}; (f) The calculated K⁺ diffusivity.

After 1st cycle activation process, the impedance of K||Cu cells exhibits an obvious decrease both in 0.4 and 1 M KPF₆-DME at -40°C. Considering the interface reaction limited by the low temperature, we then increase cycle number to 5 cycles to form a SEI with relatively stable composition. Based on the fitted results of 5th cycle, 0.4 M KPF₆-DME exhibits a lower charge-transfer resistance (Rct) of 759 Ω compared with 1 M KPF₆-DME (Rct=1638 Ω), demonstrating the decreased energy barrier for K⁺ ion-desolvation process.

The K⁺ diffusivity is calculated by the equation (*Electrochimica Acta* **53**, 5071–5075 (2008)):

$$D_{K^+} = \frac{2R^2T^2}{n^4F^4\sigma^2A^2C^2}$$

, where R , T , A , F , n , σ and C are the gas constant of 8.314 J K⁻¹ mol⁻¹, the absolute temperature of 233 K, the surface area of the electrode of 0.8 cm², the Faraday's constant of 96500 C mol⁻¹, the number of electrons per molecule, Warburg coefficient and the K⁺ concentration, respectively. Warburg coefficient is the slope of Z'-ω^{-1/2}.

From the above analysis, the calculated D_{K^+} is 1.37*10⁻¹¹, 2.03*10⁻¹² cm² s⁻¹ for 0.4 and 1 M KPF₆-DME, respectively, at -40°C.

4. The authors claim (line 110-111) that the cloudy electrolyte (0.4 M KPF₆-DME) was due to decomposition of DME solvent. Can the authors provide some references for that, or could this cloudy behaviour be due to the re-precipitation of KPF₆ salt over time?

If not, please provide evidence of DME decomposition with some chemical characterization of the electrolyte (without PDMS additive) after 48 h.

Reply: Thanks for the reviewer's comment. We respectfully point out the fact that we soaked K in DME solvent, rather than the 0.4 M KPF₆ – DME. Then, this cloudy behavior cannot be due to the re-precipitation of the KPF₆ salt over time. In addition, after K soaked in the pretreated anhydrous DME, the C=O stretching vibration peaks at 1605 and 1679 cm⁻¹ were clearly detected on the K surface, and it can be identified as decomposition products (CH₃COO⁻) from DME molecular (**Supplementary Figure 8, *Angew. Chem. Int. Ed.* 61, e202207018 (2022)**).

In the reversed manuscript, we added:

Supplementary Figure 8. ATR-FTIR spectra of decomposition products from DME.

The C=O stretching vibration peaks at 1605 and 1679 cm⁻¹ were clearly detected on the K surface after soaking in the anhydrous DME, which can be attributed to the decomposition product (CH₃COO⁻) from DME molecular. When PDMS added in the anhydrous DME, the intensity of C=O obviously decreased, demonstrating that PDMS can suppress the decomposition of DME on K metal surface.

5. Does the addition of PDMS change the ionic conductivity of electrolyte at ambient or low temperatures as compared to no PDMS additive (data given in Fig 3-SI is without PDMS addition) ? What is the freezing point of PDMS and can it function at -40°C?

Reply: We thank the reviewers for this good suggestion.

As shown in **Supplementary Figure 16**, the introduction of PDMS in 0.4 M KPF₆-DME electrolyte led to a slight decrease in ionic conductivity from 1.26, 0.89 S/m to 0.91, 0.69 S/m at 25 and -40°C, respectively.

The freezing point of PDMS (trimethylsiloxy terminated, M.W. 770, Alfa) is < -40°C from Material Safety Data Sheet (MSDS). Besides, its phase transition is not detected until -80°C

based on our DSC test (Supplementary Figure 15). Thus, PDMS should be a good cryogenic additive for designing low-temperature electrolytes.

In the reversed manuscript, we added:

Supplementary Figure 16. (a) The Nyquist plots of the 0.4 M KPF₆-DME with 2 vol. % PDMS (KDP) at different temperatures; (b) Temperature-dependent ionic conductivity of 0.4 M KPF₆-DME electrolyte with and without PDMS additive; (c) Temperature-dependent ionic conductivity of the KDP electrolyte with a series of concentration gradients.

Supplementary Figure 15a. The DSC tests of the PDMS additive.

The melting point of PDMS (trimethylsiloxy terminated, M.W. 770, Alfa) is below -40°C from Material Safety Data Sheet (MSDS). Besides, its phase transition is not detected until -80°C

based on the DSC test. Thus, PDMS should be a good cryogenic additive.

6. Reason/evidence should be provided as to why 2 vol. % is “optimal” quantity of PDMS as compared to 4 and 8 vol. % (Fig 8-SI)?

Reply: We thank the reviewers for this good suggestion. We have performed the long-term cycling of K||Cu cells with volume fractions of 0%, 1%, 2%, 4%, 6% and 8% PDMS. It is clearly seen that the 0.4 M KPF₆-DME with 2 vol. % PDMS additive shows excellent cycling stability (Fig. 3a and Supplementary Fig. 11). When the PDMS content increased to 4 and 8 vol. %, it can be found that the electrochemical polarization of the Cu||K batteries became more serious, because the PDMS of 770 increases the viscosity of the electrolyte and reduces the ionic conductivity of the electrolyte.

In the reversed manuscript, we added:

Fig. 3a K plating/stripping CE of K||Cu half cells in various electrolytes at 1.0 mA cm⁻² / 1 mAh cm⁻² and -40°C.

Supplementary Figure 11. K plating/stripping tests by K||Cu half cells in 0.4 M KPF₆-DME

with various PDMS contents at $1.0 \text{ mA cm}^{-2}/1 \text{ mAh cm}^{-2}$ and -40°C .

When the PDMS content increased to 4, 6 and 8 vol. %, it can be found that the electrochemical polarization of Cu||K batteries became more serious and its cycling stability was even worse, owing to the PDMS with a high molecular can increase viscosity and decrease ionic conductivity of electrolytes.

7. It's interesting to see low overpotentials with 0.4 M KDP at -40°C (fig 3f). However, can the authors compare overpotentials at 25°C and -40°C and explain the reason for low overpotentials obtained at -40°C in K/K symmetric cells?

Reply: Thanks for the reviewer's good suggestion. The plots of overpotential comparison at 25°C and -40°C have been added in **Supplementary Figure 17d**. It can be found that the overpotentials of K||K cells obtained at -40°C is only slightly higher than that of 25°C with current densities from 0.5 to 4 mA cm^{-2} , indicating that the overpotentials of the K||K cells with the KDP electrolyte is weakly influenced by temperature.

As for the low overpotentials obtained by the KDP electrolyte in K||K symmetric cells at -40°C , the reasons can be explained by the enhanced K^+ transport kinetics and stable SEI including: (1) a high ionic conductivity. Though the optimization of electrolyte concentration, we found that a low-concentration electrolyte of 0.4 M processed a stable solvation structure over a wide temperature range from 25 to -50°C and achieved high ionic conductivity at -40 and -50°C , which significantly contributes to the fast K-ion transport kinetics in the bulk electrolyte (**Fig. 1d,e**); (2) a weak solvation structure. Based on the R_{ct} data and DFT calculations, this 0.4 M KPF₆-DME electrolyte exhibited a low desolvation energy barrier at -40°C , thus achieving an enhanced charge transfer kinetics for K ion transport from the bulk electrolyte to K metal surface (**Fig. 1h**); (3) PDMS additive. By introducing Si-O-based additives in a weak-solvation low-concentration electrolyte, the in situ formed potassiophilic and robust interface enables uniform K deposition (**Fig. 2**).

In the reversed manuscript, we added:

Supplementary Figure 17. The rate performance of K||K cells with the KDP electrolyte at (a) 25°C and (b) -40°C, respectively; (c) Representative plating/stripping curves at different current densities at 25°C; (d) Comparison of overpotentials obtained from (a) and (b).

8. EIS data in Fig 4 c should be properly fitted to show the evolution of resistance due to SEI layer?

Reply: Thanks for the reviewer's good suggestion. Quantitative analysis of EIS plots by fitting the equivalent circuit model is shown in **Supplementary Figure 17** and **Table 4**. After first-cycle activation, it can be found that the electrode-electrolyte interphase resistance (R_f) and R_{ct} dropped to 225 and 337 Ω , respectively, but the bulk electrolyte resistance (R_s) is almost the same around 108 Ω . In the subsequent 5 cycles, the impedance remains stable, proving that PDMS contributes to the formation of a stable SEI on K anode.

Revised text on page 11, line 185-189:

Further, the electrochemical impedance spectra (EIS) of K||Cu cells were collected from the pristine to the fifth cycle (Fig. 4c and Supplementary Fig. 19, Table 4). After first-cycle activation, the R_{ct} dropped from original 4872 to 337 Ω , but the bulk electrolyte resistance (R_s) is almost the same around 108 Ω . In the subsequent cycles, the impedance remains stable, benefitting the uniform deposition and reversible stripping of K during cycling⁴².

Supplementary Figure 19. (a) Equivalent circuit model; (b-f) EIS tests of K||Cu cells with the KDP electrolyte at -40°C and the Fitted Nyquist plots at different cycles. The corresponding impedance data is shown in **Supplementary Table 5**.

Supplementary Table 5. The corresponding impedance data obtained from K||Cu cells with the KDP electrolyte at -40°C.

Pristine	After 1 cycle	After 2 cycles	After 3 cycles	After 5 cycles
----------	---------------	----------------	----------------	----------------

Rs (Ω)	108.1	107.1	107.8	108.2	107.7
Fitting error	0.54%	0.14%	1.43%	0.56%	1.13%
Rf (Ω)	397.2	225.7	271.9	235.1	229.1
Fitting error	14.7%	15.6%	11.2%	12.1%	8.9%
CPE1-T ($\Omega^{-1}\cdot s^n$)	6.49E-6	2.81E-5	1.76 E-4	1.41 E-4	2.41E-4
Fitting error	11.2%	25.2%	27.1%	15.6%	20.1%
CPE1-P (unitless)	0.86	0.75	1.036	0.63	0.62
Fitting error	2.7%	5.5%	6.1%	4.1%	3.8%
Rct (Ω)	4872.5	337.4	352.6	335.3	322.4
Fitting error	5.7%	7.6%	5.8%	3.4%	3.8%
W1-R (Ω)	1692	8000	3583	2233	2864
Fitting error	6.7%	50.7%	37.3%	28.1%	31.8%
W1-T (s)	2.9	218.2	623.3	210.3	213.5
Fitting error	17.3%	77.2%	87.2%	36.7%	53.1%
W1-P (unitless)	0.087	0.65	0.42	0.51	0.61
Fitting error	7.3%	0.43%	0.48%	0.71%	0.44%
CPE2-T ($\Omega^{-1}\cdot s^n$)	2.01E-5	6.16E-5	7.58E-5	1.512E-4	2.79E-4
Fitting error	2.8%	2.1%	3.5%	1.5%	1.0%
CPE2-P (unitless)	0.79	1.03	0.71	1.11	1.08
Fitting error	1.3%	0.9%	1.6%	1.0%	0.7%

9. Do the SEI components form at -40°C temperature dependent? Would the same SEI components form even at room temperature? The authors should provide relative ratios of SEI components at ambient temperature in comparison to -40°C .

Reply: Thanks for the reviewer's insight comment. The chemical composition of the SEI formed at 25°C was investigated by XPS in **Supplementary Figure 20**. The relative ratios of SEI components have been provided in **Supplementary Figure 21**, and we found that the relative ratios of SEI components are influenced by the synergistic effect of PDMS and temperature. As the addition of PDMS, the KF and -COK (285.6 eV) content in outer SEI layer increased, whereas the content of K_2CO_3 (288.6 eV) and -COOK (286.9 eV) decreased at both 25 and -40°C . Similarly, when the temperature decreases from 25 to -40°C , the C-F/P-F contents in both SEI films formed with and without PDMS obviously decreased, as the reaction kinetics of electrolyte decomposition reduces at low temperature.

Revised text on page 11, line 191-201:

It is found that the outer SEI layer formed on K anode with KDP electrolyte shows a KF-rich phase (peaks at 684.6 and 685.7 eV in the F 1s spectrum), accounting for 12.5% and 15.2% of the total (C+F) contents at 25 and -40°C , respectively, which is critical for the SEI stability. Moreover, the C signal decreases from 86.1% to 78.9% with the addition of PDMS at -40°C , demonstrating that organic species of the SEI originating from the DME decomposition significantly reduced (Fig. 4f,g and Supplementary Fig. 21). Specifically, a noticeable increase of the -COK (285.6 eV) was observed, whereas the content of K_2CO_3 (288.6 eV) and -COOK (286.9 eV) decreased. The same trends were also found in the SEI formed 25°C (Supplementary Fig. 20c,d). The results demonstrate that the PDMS layer can prevent electrons on the K surface

from continuously attacking C-O to form C=O, thus efficiently suppress the chain reaction of DME decomposition.

Supplementary Figure 20. In-depth XPS spectra of the K anode after 20 cycles at 25°C. F 1s in (a) KDP and (b) 0.4 M KPF₆-DME; C 1s in (c) KDP and (d) 0.4 M KPF₆-DME.

Supplementary Figure 21. Normalized ratios of different species in the SEI formed in different electrolytes and temperatures.

10. How did the authors reach to the conclusion that the SEI without PDMS would have cracks and covered with organics as the top layer (red)? References should be provided to support such claim.

Reply: Thanks for the reviewer's good suggestion. In the PDMS-free electrolyte, K is highly reactive and reacts with solvents, forming a SEI layer on the K surface. From the in-depth C 1s XPS spectra, the organics is mainly detected on the outer rather than the inner SEI. And this SEI layer possesses a low Derjaguin–Müller–Toporov (DMT) modulus of ~1156 MPa. In contrast, PDMS endow the SEI with outstanding mechanical strength with DMT modulus up

to 1489 MPa (**Supplementary Fig. 23**). Combined with the SEM images of K deposition layer in the PDMS-free electrolyte (**Fig. 4a**), it can be concluded that the SEI without PDMS is not optimized to accommodate large volume change during K plating/stripping, resulting in the unrecoverable cracks.

References have been added in this part.

Ref 9: Wang, C. et al. Extending the low-temperature operation of sodium metal batteries combining linear and cyclic ether-based electrolyte solutions. *Nat. Commun.* **13**, 4934 (2022).

Ref 14: Zhou, M. et al. Electrolytes and interphases in potassium ion batteries. *Adv. Mater.* **33**, e2003741 (2021).

Supplementary Figure 23. DMT modulus mappings of SEI formed on K anodes (a) the KDP electrolyte and (b) 0.4 M KPF₆-DME; (c) The corresponding average DMT modulus.

11. Did the cell fail after just 3 cycles when 1 M KPF₆ (DME) was used with Cu/KPTCDA full battery? A comparative voltage – specific capacity profile of all 3 electrolytes should be given in fig 15 – SI. Also please revise the caption since it is vague whether the data (Fig 15b – SI) given is for K/Cu half cell or Cu/KPTCDA full cell.

Reply: We apologize for the inadequate explanation of this part.

First, we would like to explain the content and logic of Fig 15-SI. Fig 15a-SI was the voltage-specific capacity profiles of our Cu||KPTCDA full cell, and Fig 15b-SI was used to describe the CE of K||Cu half cells. We ascribed the unsatisfactory capacity retention of Cu||KPTCDA full cell to the comparatively low reversibility of K plating/stripping at the anode side (CE=98.7%). However, by introducing new additives, the CE of K||Cu half cells can be further improved to 99.7%, and it will be reported in the future work. Thus, Fig 15b-SI was used to explain Fig 15a-SI.

Next, we would like to answer the reviewer's questions. The 1 M KPF₆-DME, as a baseline electrolyte, was used in Cu||K half cells, and this cell failed only after 3 cycles. And a comparative voltage-specific capacity profile of all 3 electrolytes will not be provided, because

these 3 electrolytes were used in K||Cu half cells rather than Cu||KPTCDA full cells. In addition, according to reviewer's suggestion, we have revised the figure caption in the new version (Supplementary Figure 28).

In the reversed manuscript, we revised:

Supplementary Figure 28. (a) Voltage-specific capacity profiles of Cu||KPTCDA full cells at different cycles with the KDP electrolyte (0.4 M KPF₆-DME with 2 vol. % PDMS); (b) The K plating/stripping CE of K||Cu half cells with 3 electrolytes at 1.0 mA cm⁻²/1 mAh cm⁻².

The Cu||KPTCDA batteries exhibit an initial discharge capacity of 95.1 mAh g⁻¹ at 25°C with KDP electrolyte, and the calculated energy density is 188 Wh kg⁻¹. Besides, a capacity retention of 63% is observed after 20 cycles, which can be attributed to the imperfect reversibility of K metal at the anode side (average CE=98.7%). However, by introducing new additives, the CE of K||Cu half cells can be further improved to 99.7%, and it will be reported in the future work.

12. The authors should provide details (formulae and table of values) as how they reached/calculated the energy density values of 188 Wh Kg⁻¹ and 152 Wh Kg⁻¹. Currently, it is unknown what parameters (voltage etc.) were incorporated to calculate these values.

Reply: We thank the reviewers for pointing this out. The specific energy density of Cu||KPTCDA cells is calculated based on the total loading mass of the cathode and anode instead of the current collectors. We also provide the detailed parameters in the **Supplementary Table 5**.

In the revised manuscript on page 20, line 330-334, we added:

$$E_{cathode+anode} = \frac{U_{avg} \times Q_{cell}}{m_{KPTCDA} + m_{CMC} + m_{Super P}}$$

where U_{avg} is the average cell operating voltage; Q_{cell} is the discharge capacity of the cell; $(m_{KPTCDA} + m_{CMC} + m_{Super P})$ is the total loading mass of cathode after prepotassiation; The weight of anode is 0; Current collectors is not considered in the formula.

Supplementary Table 6. The calculation of full-cell specific energy.

Parameters	Cell 1 (at -40°C)	Cell 2 (at -40°C)
------------	-------------------	-------------------

PTCDA:Super P:CMC	8:1:1	8:1:1
Total loading mass of cathode before prepotassiation/mg	9	6
Total loading mass of cathode after prepotassiation/mg	10.31	6.88
Total loading mass of anode/mg	0	0
Operating temperature/°C	-40	25
Operating voltage/V	2.39	2.41
Discharge capacity of Cu KPTCDA cells/mAh	0.655	0.538
Specific capacity of Cu KPTCDA cells/mAh g ⁻¹	77.5	95.1
Energy density based on the total loading mass of cathode and anode/Wh kg ⁻¹	152.6	188.4

13. Are the energy density values calculated on coin cell or pouch cell level? The energy density values at pouch cell level should also be provided to determine potential this system holds as claimed by the authors?

Reply: We thank the reviewers for this question. The energy density values presented in current version were calculated on coin cell. According to the reviewers' suggestion, the energy density values at pouch cell level is calculated to be 73.97 Wh kg⁻¹_{cathode+anode} (**Table R1**). Although our Cu||KPTCDA coin cells have achieved stable cycling and high energy density results, our pouch cell does not carefully consider some factors, such as cell expansion, E/C ratio, strict weight control and uniform external pressure (*Nat. Energy* **4**, 551–559 (2019)). Thus, extending the cycling life and realizing a high energy density in a pouch cell configuration become the top priorities in the future work.

Table R1. The calculated energy density of Cu||KPTCDA pouch cell.

Parameters	Pouch Cell
PTCDA:Super P:CMC	8:1:1
Total loading mass of cathode before prepotassiation/mg	183.8

Total loading mass of cathode after prepotassiation/mg	200.6
Total loading mass of anode/mg	0
Operating temperature/°C	-40
Operating voltage/V	2.36
Discharge capacity of Cu KPTCDA cells/mAh	6.288
Energy density based on the total loading mass of cathode and anode/Wh kg ⁻¹	73.97

14. How these energy density values compared with anode free Li and Na based batteries (at low temperatures) to validate the need to develop K anode free batteries.

Reply: We thank the reviewers for this question. As far as we know, there are no reports of anode-free lithium and sodium-based batteries at low temperatures. In the Introduction part, we found that the reported CE of alkali metal anodes (Li, Na and K) is far from satisfactory and never exceeding 99% below -20°C, which seriously hinder the development of low-temperature anode-free batteries. Then, compared with Li⁺ and Na⁺ ions, K⁺ ions have weak Coulombic interactions with solvent molecules and would exhibit the small Stokes radii, which can effectively reduce the ion diffusion resistance in electrolytes. Based on the above analysis, the possibility of a low-temperature anode-free K metal battery is further explored. By introducing Si-O-based additives in a weak-solvation low-concentration electrolyte, we realize a low-temperature anode-free K metal battery, which represents a critical step toward the basic scientific research of low-temperature anode-free metal batteries.

15. The authors have shown low cycle no. (50 cycles in coin cell) for K anode-free battery. What do the authors think about this low cycle no. performance?

Reply: We agree with reviewer's comment that the cycle number of our K anode-free batteries was relatively low. Considering that the capacity retention of the battery should be >80%, we put the first 50 cycles in Fig.6c. However, as a proof of concept, this cycling performance (an 82% capacity retention after 50 cycles and a 70% capacity retention after 92 cycles at -40°C) is still acceptable (Figure. R5), given the great potential of performance improvement by further optimizations of materials processing and battery assembly. Besides, through literature screening, we found that only 1 research article was reported with a slight excess of alkali metal anodes (≤onefold excess) at -40°C, and our Cu||KPTCDA cells exhibited a better cycling stability than the reported 1×Li || sulfurized polyacrylonitrile full cells at -40°C (a 63% capacity retention after 50 cycles, *Nat. Energy* 6, 303–313 (2021)). Moreover, the cycle performance of our low-temperature Cu||KPTCDA cells is also comparable to the previously reported room-

temperature anode-free batteries including Li/Na/K chemistries (Table R2).

Figure. R5. Cycling performance of the Cu||KPTCDA full battery at 0.2C and -40°C.

Table R2. Summary of cycle performances of anode-free coin batteries.

Battery configuration	working temperature (°C)	Cycle number	Capacity retention (%)	Ref.
Cu KPTCDA	-40	50 92	82 70	This work
1× Li sulfurized polyacrylonitrile	-40	50	63	Nat. Energy 6 , 303–313 (2021)
Al@graphene K-FeS ₂	25	30	32	Adv. Mater. 34 , 2202902 (2022)
Cu LiFePO ₄	25	100	54	Adv. Funct. Mater. 26 , 7094–7102 (2016)
Al/Graphitic carbon Na[Cu _{1/9} Ni _{2/9} Fe _{1/3} Mn _{1/3}]O ₂	25	200	82	Nat. Energy 7 , 511–519 (2022)
Al O3-NaCu _{1/9} Ni _{2/9} Fe _{1/3} Mn _{1/3} O ₂	25	250	73.1	Angew. Chem. Int. Ed. 61 , e202200410 (2022)

16. The electrolyte used is a dilute KPF₆ would run out quickly (as compared to high molar ratios) during cycling of anode-free batteries. Therefore, what is the significance of using low conc. electrolyte in this work, irrespective of its functionality at low temperatures?

Reply: We thank the reviewer for this insightful comment. In fact, whether the electrolyte run quickly depends on the compatibility issues between electrodes and electrolytes. Moreover, due to the lack of suitable electrolytes, anode-free cells are being hindered by low reversibility, unstable SEI and uncontrolled dendrite growth of alkali metal at the anode side. In this work, we have greatly improved the reversibility of K metal plating/stripping to achieve a record-high CE of 99.80% by electrolyte regulation, demonstrating a low consumption rate of our electrolyte.

As regards the significance of using low concentration electrolyte, there are some other advantages: (1) The low-concentration electrolyte can reduce the cost (*Nat. Energy*, **4**, 269–280 (2019)); (2) It exhibits low viscosity and is more likely to mitigate the negative effect of low

temperature on ionic conductivity (*Adv. Funct. Mater.* **32**, 2205393 (2022)); (3) It can reduce corrosive risk of correctors (less HF from PF₆⁻ decomposition), which would help improve the battery's stability (*ACS Energy Lett.* **5**, 1156-1158 (2020)). (4) Low-concentration electrolytes generally exhibit a weak solvation and a low charge transfer impedance (*Angew. Chem. Int. Ed.* **61**, e202215866 (2022)).

More importantly, low-concentration electrolytes have also been widely reported in recent literature including Li/Na/K batteries (*ACS Energy Lett.* **5**, 1156-1158 (2020); *Angew. Chem. Int. Ed.* **61**, e202213416 (2022); *Angew. Chem. Int. Ed.* **61**, e202215866 (2022); *Adv. Funct. Mater.* **32**, 2205393 (2022); *Adv. Mater.* **34**, 2205678 (2022)).

17. Overall language of the manuscript could be improved for better understanding. The current version lacks proper sentence structure.

Reply: We are grateful for the reviewer's comment. We have carefully checked and improved the English writing in the revised manuscript, and we have also applied Nature Research Editing Service, a formal language polishing organization, to polish English language and standardize sentence structure of this manuscript. The English Editing Certificate can be found here:

Reviewer #3 (Remarks to the Author):

It is an interesting study on low temperature behaviors of potassium (K) battery chemistry. Nevertheless, the significance of the study needs improvement as the temperature is mostly limited at -40°C (A phase transition, which might be due to salt precipitation, was observed in differential scanning calorimetry, indicating the temperature limitation of formulated electrolyte is around -50°C). The state of the art studies demonstrates low temperature performance of alkali metal batteries down to -85°C, such as DOI: 10.1002/anie.201900266 and doi.org/10.1038/s41560-019-0474-3. Additionally, some mechanisms are not well explained and some critical experimental/analytical details are missing in the current manuscript.

Reply: Thanks very much for the reviewer's insightful comments.

Just as the reviewer mentioned, this temperature interval is mainly limited by the high freezing point DME around -58°C . So, we carried out the electrochemical performance of $\text{K}||\text{K}$ and $\text{Cu}||\text{KPTCDA}$ cells over a temperature range from 25 to -50°C . Although our work cannot extend the temperature range to -85°C , this is the first time that the working temperature range of anode-free cells has been extended to -40°C . The notable characteristic distinguished from the reported low-temperature metal batteries is the completely avoided overuse of metal anodes, thus promising to improve the energy density at cell level. As far as we know, there are no reports of K-ion full cells (overusing K metal anode is not included) at -40°C , and the energy density (calculated based on the total loading mass of cathode and anode) of our $\text{Cu}||\text{KPTCDA}$ cells is very comparable to Li/Na-ion full cells at -40°C (**Supplementary Figure 29, Table 7**). The successful implementation of our work provides a potential strategy to realize the low-temperature high-energy batteries. Certainly, further progress in achieving anode-free batteries at a lower operating temperature will be carried out in future work. Moreover, additional mechanism studies and experimental/analytical details are also provided in the revised manuscript. We hope that this version is suitable for publication in Nature communications.

Supplementary Figure 29. Comparison of energy densities with representative Li/Na-ion full cells at -40°C . The energy densities were calculated based on the total loading mass of cathode and anode. Note: the data listed in **Supplementary Table 7** is used for the plot.

Supplementary Table 7. Comparisons of the electrochemical performance between this work and the reported low-temperature rechargeable batteries.

Cell Configuration	Operating Temperature ($^{\circ}\text{C}$)	Current rate (A g^{-1})	Average discharge voltage (V)	Specific energy (Wh kg^{-1})	Ref.
$\text{Cu} \text{KPTCDA}$	-40	0.026	2.39	152	This work
$\text{Bi} \text{Na}_4\text{Fe}_3(\text{PO}_4)_2\text{P}_2\text{O}_7$	-40	0.02	2.3	127.5	Angew. Chem. Int. Ed. 61, e202116930 (2022)

HC NFPF@C@MCNTs	-40	0.037	2.1	39.4	Small 18 , 2204830 (2022)
HC P2-NaMNNb	-40	0.037	3.2	188	Nat. Commun. 13 , 3205 (2022)
AG PTPAn	-40	0.01	2.5	118	Angew. Chem. Int. Ed. 60 , 23858–23862 (2021)
NC LiNi _{0.65} Co _{0.15} Mn _{0.2} O ₂	-40	0.012	2.44	146.8	Angew. Chem. Int. Ed. 61 , e202209619 (2022)
Gr NMC811	-40	0.02	3.4	129	Angew. Chem. Int. Ed. 134 , e202205967 (2022)
Li _x C Li ₂ V ₂ (PO ₄) ₃	-40	0.017	3.92	139	Angew. Chem. Int. Ed. 56 , 16606–16610 (2017)

Note: the energy density is calculated based on the total loading mass of cathode and anode.

1. The average coulomb efficiency (CE) of KPF₆-DME system is lower than that of KFSI-EC/DEC in supplementary Fig.1. Why it is selected for low temperature study? What is more, all the samples show irregular morphology of deposition on current collector (supplementary Fig. 2). What is the logic to choose KPF₆-DME over others? Besides, the screening was performed at room temperature, which is not relevant to low temperature investigation. Low temperature screening is required.

Reply: We thank the reviewer for this professional comment. In fact, K||Cu cells in KPF₆-DME electrolyte exhibited the highest initial CE and the lowest overpotential among 4 representative electrolytes. Owing to the internal short-circuit, it seemed that the CE of KPF₆-DME system was lower than that of KFSI-EC/DEC after 3 cycles (**Figure R6**). Interestingly, the SEI stability and average CE of KPF₆-DME electrolyte can be largely improved by the PDMS additive. According to the review's suggestion, the low-temperature electrochemical behavior of K||Cu cells with these 4 electrolytes has been added in the revised manuscript. As shown in **Supplementary Figure 3**, with the temperature decreasing to -20°C, the K||Cu with 0.8 M KPF₆-EC/DEC, 1 M KFSI-EC/DEC and 3 M KFSI-DME electrolytes does not work. In contrast, the K||Cu cell with 1 M KPF₆-DME can still plating with a low overpotential. From the above analysis, the KPF₆-DME electrolyte with PDMS additive is a reasonable choice for the low-temperature K-ion batteries.

Figure R6. The K plating/stripping test in K||Cu cells with 1 M KPF₆-DME.

In the revised manuscript, we added:

Supplementary Figure 1. (a) CE tests of the K||Cu half cells cycled with different electrolytes at 1.0 mA cm⁻²/1 mAh cm⁻² (25°C); (b) The corresponding voltage profiles of K plating/stripping in the first cycle; Voltage profiles of K plating on Cu current collectors at (c) 0°C and (d) -20°C after resting for 3 h.

It can be found that K||Cu half cells with 1 M KPF₆-DME exhibits the highest initial CE and lowest electrochemical polarization, but a poor cycle life of three cycles at 25°C. With the temperature decreased to 0 and -20°C, the K||Cu half cells only with 1 M KPF₆-DME electrolyte can work with a small overpotential during the plating process.

Revised text on page 4, line 75-77:

Conventional electrolyte systems for K⁺ ion battery¹⁴⁻¹⁸ including 0.8 M KPF₆-EC/DEC, 1 M KPF₆-DME, 1 M KFSI-EC/DEC and 3 M KFSI-DME were screened, and the KPF₆/DME-based electrolyte system exhibits relatively high compatibility with K metal, in terms of higher initial CE, lower nucleation overpotential and well cold-adapted property (Fig. 1c and

Supplementary Figures. 1,2).

2. In supplementary Fig. 5, EIS fitting and equivalent circuit are required to determine the charge-transfer resistance. Why the EIS tests were conducted after 5 cycles? What are the EIS profiles at 1st cycle?

Reply: We thank the reviewers for this question. EIS fitting and equivalent circuit have been added in **Supplementary Figure 6**. Considering the interface reaction limited by the low temperature, we then increase cycle number to 5 cycles to form a SEI with relatively stable composition for the EIS analysis. It can be found that the 0.4 M KPF₆-DME exhibits a lower charge-transfer resistance (R_{ct}) of 759 Ω compared with 1 M KPF₆-DME (R_{ct}=1638 Ω).

The EIS profiles at 1st cycle are also added in **Supplementary Figures. 6a,b**, which exhibits a higher impedance than that of the 5th cycles for both 0.4 and 1 M KPF₆-DME. Consistent with present conclusion, the 0.4 M KPF₆-DME exhibits a lower R_{ct} compared with that of 1 M KPF₆-DME (**Figure R7**).

Figure R7. (a) Fitted Nyquist plots at 1st cycle; (b) The corresponding impedance data.

In the revised manuscript, we added:

Supplementary Figure 6. EIS tests of K||Cu cells at different cycles in (a) 1 M and (b) 0.4 M KPF₆-DME, respectively, at -40°C; (c) Fitted Nyquist plots at 5th cycle; (d) Equivalent circuit model and the corresponding impedance data; (e) The plot of $Z' - \omega^{-1/2}$; (f) The calculated K⁺ diffusivity.

After 1st cycle activation process, the impedance of K||Cu cells exhibits an obvious decrease both in 0.4 and 1 M KPF₆-DME at -40°C. Considering the interface reaction limited by the low temperature, we then increase cycle number to 5 cycles to form a SEI with relatively stable composition. Based on the fitted results of 5th cycle, 0.4 M KPF₆-DME exhibits a low charge-transfer resistance (R_{ct}) of 759 Ω compared with 1 M KPF₆-DME (R_{ct} =1638 Ω), demonstrating the decreased energy barrier for K⁺ ion-desolvation process.

3.The authors claimed that DME can react with K metal (line 108 and 109 on page 6). Then why the K metal was used to pretreat DME solvent (line 253 on page 15)?

Reply: Thanks for the reviewer's insightful comments. There are two key points that explain this: (1) K can react with DME, but K first reacts with the trace amount of water in DME. The reaction process is accompanied by large number of bubbles and yellow flocculent precipitates (Supplementary Figure 7); (2) After pretreatment to remove trace residual water, DME can

still react with K, and the decomposition product of DME molecular on K metal surface was clearly detected by ATR-FTIR and it could be identified as CH_3COO^- (**Supplementary Figure 8**, *Angew. Chem. Int. Ed.* **61**, e202207018 (2022)). Hence, the PDMS additive was introduced in the KPF_6/DME electrolyte to suppress the continuous decomposition of DME on K metal surface. The control experiments (**Fig.2a**) and enhanced electrochemical performance can further verify this.

In the revised manuscript, we added:

Supplementary Figure 7. Digital photographs of reaction phenomena between K and DME.

When K metal was soaked in purchased DME (Sigma-aldrich, anhydrous, 99.5%, inhibitor-free), an obvious chemical reaction between K and H_2O occurred, accompanied by large number of bubbles and yellow flocculent precipitates. After reaction, the completely anhydrous DME was obtained by collecting the supernatant liquid. Then, we repeated the above-mentioned tests, and found that DME became more stable with K metal, which means that K mainly reacts with water rather than DME. Thus, it is a logical treatment to remove the residual water in DME by K metal.

Supplementary Figure 8. ATR-FTIR spectra of decomposition products from DME.

After removal of water by K metal, the obtained anhydrous DME can still react with K. From the ATR-FTIR spectra, the C=O stretching vibration peaks at 1605 and 1679 cm^{-1} were clearly detected on the K surface, which can be attributed to the decomposition product (CH_3COO^-) from DME molecular. When PDMS added in the anhydrous DME, the intensity of C=O obviously decreased, demonstrating that PDMS can suppress the decomposition of DME on K metal surface.

4. In supplementary Fig. 8, the authors stated that 2 vol % is the optimized additive concentration. However, no values less than 2 vol % were investigated. The authors are required to provide supporting results.

Reply: We thank the reviewers for this good suggestion. We have added the CE tests of K||Cu cells with 0% and 1% PDMS in Fig. 3a and Supplementary Figure 11, and 0.4 M KPF₆-DME with 2 vol. % PDMS additive shows excellent K plating/stripping behavior.

In the revised manuscript, we added:

Fig. 3a K plating/stripping CE of K||Cu half cells in various electrolytes at 1.0 mA cm^{-2} / 1 mAh cm^{-2} and -40°C .

Supplementary Figure 11. K plating/stripping tests by K||Cu half cells in 0.4 M KPF₆-DME with various PDMS contents. electrolytes at 1.0 mA cm⁻²/1 mAh cm⁻² and -40°C.

When the PDMS content increased to 4, 6 and 8 vol. %, it can be found that the electrochemical polarization of Cu||K batteries became more serious and its cycling stability was even worse, owing to the PDMS with a high molecular can increase viscosity and decrease ionic conductivity of electrolytes.

5. In Fig. 3a, the authors need to reveal the initial 20 cycles of CE in the figure inset. For Fig. 3e, the authors are required to provide the corresponding rate capability test as a plot of capacity vs cycle with CE. What is the recovery rate when current density goes back to 0.5 mA cm⁻²?

Reply: We thank the reviewers for this good suggestion. The CE values of the initial 20 cycles have been provided in the **Supplementary Figure 11**. The initial CE of K||Cu cells with the KDP electrolyte is 97.19%. After only one cycle, the CE is higher than 99% and then increases to 99.82% after 15 cycles.

Besides, the rate capacity test of K||Cu cells at -40°C has been added in **Supplementary Figure 13**, and when current density goes back to 0.5 mA cm⁻², the recovery rate is 99.6%. Also, the rate capacity test of K||Cu cells at 25°C has been provided in **Figure R8**. The CE values are 98.65%, 98.82%, 98.63%, 98.62%, 98.48%, 98.58% and 99.96% at 0.5, 1, 2, 3, 4, 6 and 8 mA cm⁻², respectively. When the current density was brought back to 0.5 mA cm⁻², the CE is 98.54%, and the recovery rate is 99.8%.

Revised text on page 9, line 146-151:

Besides, the reversibility of K plating/stripping process over a wide range of current densities is also investigated in K||Cu cells at -40°C. An average CE of 99.84% with stable polarization is observed at 0.5 mA cm⁻². The cell cycles well at higher current density of 2 mA cm⁻², demonstrating a stable CE of 98.95%. When brought back to 0.5 mA cm⁻², K||Cu cells with a CE marginally higher than 100% is observed at the beginning of cycles, indicating recovery of dead K formed at higher current densities (Supplementary Fig. 13)⁶.

Supplementary Figure 11. The enlarged plot of CE values with 2 vol. % PDMS at -40°C .

Supplementary Figure 13. The rate capability test of K||Cu cells at 1 mAh cm^{-2} and -40°C .

Figure R8. The rate capability test of K||Cu cells at 1 mAh cm^{-2} and 25°C .

6. In supplementary Fig. 10, the surfaces of K metals using the formulated electrolyte are not uniform but porous. The author needs to provide explanation.

Reply: Thanks for reviewing our manuscript carefully. We would like to note that the surfaces of K metals using the formulated electrolyte is homogenous and smooth. To clearly this point,

the SEM images of K anodes after 10 cycles at -40°C in the KDP and PDMS-free $0.4\text{ M KPF}_6\text{-DME}$ electrolyte has been provided for comparison, and large porous was observed in the PDMS-free electrolyte (**Supplementary Figure 18a,b**). In a sharp contrast, the KDP electrolyte favored a homogenous and smooth K surface morphology (**Supplementary Figure 18c,d**).

In supplementary Fig. 10 of our original version, we provided the SEM images of K after 100 cycles at -40°C , and they exhibited relatively homogenous and smooth morphologies (**Supplementary Figure 18e,f**).

Overall, the KDP electrolyte yielded a uniform K metal deposition layer at -40°C , confirming that PDMS contributes to the interfacial stability and serve as a potassiophilic interface to guide uniform K deposition.

In the revised manuscript, we added:

Supplementary Figure 18. SEM images of K metal anode after 10 cycles at -40°C : $0.4\text{ M KPF}_6\text{-DME}$ electrolyte at (a) $1\text{ mA cm}^{-2}/1\text{ mAh cm}^{-2}$ and (b) $2\text{ mA cm}^{-2}/1\text{ mAh cm}^{-2}$; the KDP electrolyte at (c) $1\text{ mA cm}^{-2}/1\text{ mAh cm}^{-2}$ and (d) $2\text{ mA cm}^{-2}/1\text{ mAh cm}^{-2}$. SEM images of K metal anode after 100 cycles with the KDP electrolyte at (e) $1\text{ mA cm}^{-2}/1\text{ mAh cm}^{-2}$ and (f) $2\text{ mA cm}^{-2}/1\text{ mAh cm}^{-2}$, respectively (-40°C).

After 10 cycles at -40°C , the K||K cells were then disassembled to observe the morphology of the K anodes. When $0.4\text{ M KPF}_6\text{-DME}$ electrolyte was used, extremely porous K was observed (**Supplementary Figure 18a,b**). In contrast, the PDMS system yielded smooth K metal surface (**Supplementary Figure 18c,d**). Even after 100 cycles, the relatively uniform K surface morphologies were still observed, confirming that the PDMS can be conducive to improve the interfacial stability and serve as a potassiophilic interface to guide uniform K deposition (**Supplementary Figure 18e,f**).

7. In Fig. 4c, both equivalent circuit and electrochemical impedance spectroscopy (EIS) fitting are needed. The authors need to provide quantitative analysis.

Reply: We thank the reviewers for this good suggestion. Both equivalent circuit and electrochemical impedance spectroscopy (EIS) fitting have been provided in **Fig. 4c, Supplementary Figure 19 and Table 4**. After first-cycle activation, the R_{ct} dropped from 4872 Ω , but the bulk electrolyte resistance (R_s) is almost the same around 108 Ω . In the subsequent cycles, the impedance remains stable, which facilitates the uniform plating and reversible stripping of K during cycling.

Revised text on page 11, line 185-189:

Further, K||Cu cells were used to reveal the electrochemical impedance build-up from the pristine to the fifth cycle (Fig. 4c and Supplementary Fig. 19, Table 4). After first-cycle activation, the R_{ct} dropped from 4872 to 337 Ω , but the bulk electrolyte resistance (R_s) is almost the same around 108 Ω . In the subsequent cycles, the impedance remains stable, benefitting the uniform plating and reversible stripping of K during cycling. The results further proved that PDMS contributes to the formation of a stable SEI on K anode.

Fig.4c Nyquist plots of the K||Cu cells cycled at -40°C in KDP electrolyte.

Supplementary Figure 19. (a) Equivalent circuit model; (b-f) EIS tests of K||Cu cells with the KDP electrolyte at -40°C and the Fitted Nyquist plots at different cycles. The corresponding impedance data is shown in **Supplementary Table 5**.

Supplementary Table 5. The corresponding impedance data obtained from K||Cu cells with the KDP electrolyte at -40°C .

	Pristine	After 1 cycle	After 2 cycles	After 3 cycles	After 5 cycles
R_s (Ω)	108.1	107.1	107.8	108.2	107.7
Fitting error	0.54%	0.14%	1.43%	0.56%	1.13%
R_f (Ω)	397.2	225.7	271.9	235.1	229.1
Fitting error	14.7%	15.6%	11.2%	12.1%	8.9%
$CPE1-T$ ($\Omega^{-1}\cdot s^d$)	6.49E-6	2.81E-5	1.76 E-4	1.41 E-4	2.41E-4
Fitting error	11.2%	25.2%	27.1%	15.6%	20.1%
$CPE1-P$ (unitless)	0.86	0.75	1.036	0.63	0.62
Fitting error	2.7%	5.5%	6.1%	4.1%	3.8%
R_{ct} (Ω)	4872.5	337.4	352.6	335.3	322.4
Fitting error	5.7%	7.6%	5.8%	3.4%	3.8%
$W1-R$ (Ω)	1692	8000	3583	2233	2864
Fitting error	6.7%	50.7%	37.3%	28.1%	31.8%
$W1-T$ (s)	2.9	218.2	623.3	210.3	213.5
Fitting error	17.3%	77.2%	87.2%	36.7%	53.1%
$W1-P$ (unitless)	0.087	0.65	0.42	0.51	0.61
Fitting error	7.3%	0.43%	0.48%	0.71%	0.44%
$CPE2-T$ ($\Omega^{-1}\cdot s^d$)	2.01E-5	6.16E-5	7.58E-5	1.512E-4	2.79E-4
Fitting error	2.8%	2.1%	3.5%	1.5%	1.0%
$CPE2-P$ (unitless)	0.79	1.03	0.71	1.11	1.08
Fitting error	1.3%	0.9%	1.6%	1.0%	0.7%

8. The formation and protection mechanism of organic-inorganic hybrid interphase (line 187 on page 10) are unclear. How are the mechanical properties of the interphase? More relevant characterization tools (i.e. AFM) are recommended.

Reply: We thank the reviewers for this good suggestion. AFM characterization has been performed as shown in **Supplementary Figure 23**, and the SEI formed in the PDMS-free electrolyte possesses a Derjaguin–Müller–Toporov (DMT) modulus of ~1156 MPa. In contrast, the PDMS additives contribute to a robust SEI with a higher DMT modulus of ~1489 MPa. From XPS and Raman mapping tests, we found a complete and uniform SEI with increased fractions of KF and Si–O–based components formed and uniformly distributed on the K metal surface. Furthermore, based on the DFT calculations and SEM images of K deposition morphology, it can be concluded that this SEI with PDMS is optimized to accommodate large volume change during K plating/stripping, resulting in a smooth and uniform K surface morphologies. The high CE and stable cycling of K anodes also indicate the formation of a protective SEI. From the above analyses, this organic-inorganic hybrid interphase has strong mechanical strength, which can buffer the volume variations and suppress the growth of K dendrite during cycling.

Revised text on page 14, line 226-232:

To probe the mechanical properties of SEI on the K anode, atomic force microscopy (AFM) was conducted. The SEI formed in the PDMS-free electrolyte possesses a Derjaguin–Müller–Toporov (DMT) modulus of ~1156 MPa (Supplementary Fig. 23a). In contrast, Si–O–Si linkages endow the SEI with outstanding mechanical strength, resulting in a significant increase of the DMT modulus up to 1489 MPa, which is mechanically strong to buffer the volume variations and suppress K dendrite growth during cycling (Supplementary Fig. 23b,c).

Supplementary Figure 23. DMT modulus mappings of SEI formed on K anodes (a) the KDP electrolyte and (b) 0.4 M KPF₆-DME; (c) The corresponding average DMT modulus.

9. DSC profile (supplementary Fig. 3a) is not complete, both cooling and heating curves need to be revealed.

Reply: Thanks for the reviewer's good suggestion. DSC profiles with both cooling and heating curves has been provided in **Supplementary Figure 15**. Based on the analysis of exothermic peak occurred from the cooling curves, the phase transition temperature of the KDP electrolyte is around -54°C .

Revised text on page 9, line 155-158:

Moreover, K||K cells with the KDP electrolyte can stably cycle over the temperature range of 25°C to -50°C , and the sudden increase of the overvoltage at -60°C is due to solidification of the electrolyte, which is identified by the observable endothermic peak arising around -54°C through differential scanning calorimetry (DSC) (Supplementary Fig. 15).

Supplementary Figure 15. The DSC tests of (a) PDMS and (b) the KDP electrolyte.

The melting point of PDMS (trimethylsiloxy terminated, M.W. 770, Alfa) is $< -40^{\circ}\text{C}$ from Material Safety Data Sheet (MSDS), Besides, its phase transition is not detected until -80°C based on the DSC test. Thus, PDMS should be a good cryogenic additive.

10. Temperature dependent ionic conductivity of formulated electrolyte (with additive) need to be presented and compared.

Reply: We thank the reviewers for this good suggestion. This part has been added in the revised manuscript. As shown in **Supplementary Figure 12**, the introduction of PDMS in 0.4 M KPF₆-DME electrolyte resulted in a slight decrease in ionic conductivity from 0.89, 0.80 S/m to 0.69, 0.61 S/m at -40 and -50°C , respectively. Moreover, 0.4 M electrolyte still achieve the highest ionic conductivity at -50°C among a series of concentration gradients.

In the revised manuscript, we added:

Supplementary Figure 12. (a) The Nyquist plots of the 0.4 M KPF₆-DME with 2 vol. % PDMS at different temperatures; (b) Temperature dependent ionic conductivity of 0.4 M KPF₆-DME electrolyte with and without PDMS additive; (c) Temperature-dependent ionic conductivity of KPF₆-DME with 2 vol. % PDMS electrolytes with a series of concentration gradients.

11. The additive amounts vary in K||Cu (K||K) cells and Cu||KPTCDA cells. For K||Cu and K||K cells, 60 μL electrolyte (1.2 μL additive) is added. While for Cu||KPTCDA cells, 100 μL electrolyte (2 μL additive) is used for coin cells and 3000 μL (60 μL additive) for pouch cells. What is the logic to use different electrolyte amounts?

Reply: Thanks for the reviewer's insightful comments.

Regarding different electrolyte volumes, we provide some explanations as follow: (1) For K||K and K||Cu cells, only surface wetting of K metal and Cu current collector is required. Whereas, the KPTCDA cathode used in the anode-free battery is designed with a high mass loading of $\sim 6.5 \text{ mg cm}^{-2}$, and requires a relatively high amount of electrolyte to ensure sufficient wetting the KPTCDA cathode; (2) Although different amounts of electrolyte are used in coin and pouch cells, the design amount is set at around 200 $\mu\text{L}/\text{mAh}$; (3) The additive volume fraction remains the same as 2%, maintaining the physicochemical property (i.e. ionic conductivity, viscosity and solvation structure) of our electrolyte unchanged.

12. In line 223 on page 13, the authors discussed that the cell shows high safety. Nonetheless, no supporting results are provided. The authors need to provide supportive results (i.e. DSC analysis).

Reply: Thanks for the reviewer's question.

The anode-free full-cell architecture is constructed with a cathode and a bare Cu current collector, but does not use alkali metal foil. Thus, the anode-free cell architecture exhibits advantages in terms of cost, safety and cell assembly procedure compared with a typical alkali metal battery.

Revised text on page 16, line 260-263:

Owing to the high-safe cell architecture (without metal foil anodes) and easy assembly procedure of anode-free cells compared with conventional alkali metal batteries, a Cu||KPTCDA anode-free pouch cell (5 cm × 7 cm) was successfully constructed (Fig. 6e).

13. For Fig. 6g, the authors need to provide both charge capacity and CE. What is more, temperature dependent cyclic voltammetry and galvanostatic charge-discharge profile are need for Cu||KPTCDA cells in Fig. 6.

Reply: Thanks for the reviewer's constructive comment.

Both charge capacity and CE have been provided in **Fig. 6g**, and the CE of the Cu||KPTCDA pouch cell is gradually close to 100% after first-cycle activation process. Besides, the temperature dependent cyclic voltammetry and galvanostatic charge-discharge profile of Cu||KPTCDA cells have also been added shown in **Fig. 6b** and **Supplementary Figure 26**. It can be found that CV curves exhibit three oxidation/reduction peaks at around 3.2/2.25, 2.99/2.58, and 2.79/2.84V, accompanied by a slight increase in polarization as the temperature decreased from 25 to -50°C. Moreover, Cu||KPTCDA cells with the KDP electrolyte show an initial capacity of 98.6 mAh g⁻¹ at 25°C. When the temperature decreased to -40°C, the reversible capacity is 82.8 mAh g⁻¹ with a retention rate of 84%. Even at -50°C, it can still maintain a high capacity retention of 64%, well demonstrating the considerably advanced low-operating temperature performance (Fig. 6b).

Revised text on page 16, line 247-253:

As shown in Supplementary Fig. 26, the cyclic voltammetry (CV) curves exhibit three oxidation/reduction peaks at around 3.2/2.25, 2.99/2.58, and 2.79/2.84V, accompanied by a slight increase of the polarization with temperature decreased from 25 to -50°C, indicating a good redox ability at low temperature. Moreover, Cu||KPTCDA cells with the KDP electrolyte show a high capacity of 98.6 mAh g⁻¹ at 25°C. When the temperature decreased to -40°C, a reversible capacity of 82.8 mAh g⁻¹ can be achieved with a retention of 84%. Even at -50°C, a high capacity retention of 64% is still maintained, well demonstrating the considerably advanced low-operating temperature performance (Fig. 6b).

Fig. 6g Electrochemical performance of the anode-free Cu||KPTCDA pouch cell at 10 mA g⁻¹ and -40 °C.

Supplementary Figure 26. CV curves of Cu||KPTCDA cells over a temperature range from 25 to -40 °C.

Fig. 6b Voltage profiles of Cu||KPTCDA at different temperature.

REVIEWERS' COMMENTS

Reviewer #1 (Remarks to the Author):

The revised version addresses all the comments and Questions the original manuscript raised.

I do recommend the manuscript for publication in Nature Communications.

Reviewer #2 (Remarks to the Author):

The response to revisions requested is well carried out and the work is suitable for publication